# A numerical investigation on the energetics of a current along an ice-covered continental slope

Hengling Leng[1], Hailun He[1], Michael A. Spall[2]

[1]State Key Laboratory of Satellite Ocean Environment Dynamics, Second Institute of Oceanography, Ministry of Natural Resources, Hangzhou 310012, China
[2]Woods Hole Oceanographic Institution, Woods Hole, MA 02543, USA

*Correspondence to*: Hengling Leng (hengling_leng@hotmail.com); Hailun He (hehailun@sio.org.cn)

**Abstract.** The Chukchi Slope Current is a westward-flowing current along the Chukchi slope, which carries Pacific-origin water from the Chukchi shelf into the Canada Basin and helps set the regional hydrographic structure and ecosystem. Using a set of experiments with an idealized primitive equation numerical model, we investigate the energetics of the slope current during the ice-covered period. Numerical calculations show that the growth of surface eddies is suppressed by the ice friction, while perturbations at mid-depths can grow into eddies, consistent with linear instability analysis. However, because the ice stress is spatially variable, it is able to drive Ekman pumping to decrease the available potential energy (APE) and kinetic energy of both the mean flow and mesoscale eddies over a vertical scale of 100 m, well outside the frictional Ekman layer. The rate at which the APE changes is determined by the vertical buoyancy flux, which is negative as the ice-induced Ekman pumping advects lighter (denser) water upward (downward). A scaling analysis shows that Ekman pumping will dominate the release of APE for large scale flows, but the effect of baroclinic instability is also important when the horizontal scale of the mean flow is the baroclinic deformation radius and the eddy velocity is comparable to the mean flow velocity. Our numerical results highlight the importance of ice friction in the energetics of the slope current and eddies, and this may be relevant to other ice-covered regions.

## 1 Introduction

Continental slopes of Arctic shelf seas are usually featured with strong boundary currents, which transport waters and nutrients and play a crucial role in shaping the Arctic ecosystem (Bluhm et al., 2020; Oziel et al., 2022). In the western Arctic Ocean, the Chukchi continental slope is known to be the main passage for the Pacific-origin water after it exits the Chukchi shelf through Barrow Canyon (e.g., Corlett and Pickart, 2017; Li et al., 2019; Stabeno and McCabe, 2020) (see the schematic circulation in Fig. 1). The westward-flowing current along the Chukchi slope, named the Chukchi Slope Current (CSC) (Corlett and Pickart, 2017), is suggested to be distinct from the southern arm of the Beaufort Gyre (BG) due to their different origins (Watanabe et al., 2017; Spall et al., 2018). Onshore of the slope current is the bottom-intensified, eastward-flowing Chukchi Shelfbreak Jet (Corlett and Pickart, 2017; Li et al., 2019). In addition, there is an eastward-flowing current overlying the mid-

slope of the Chukchi Sea, known as the inshore branch of the Atlantic Water Boundary Current (Corlett and Pickart, 2017; Stabeno and McCabe, 2020; Li et al., 2020).

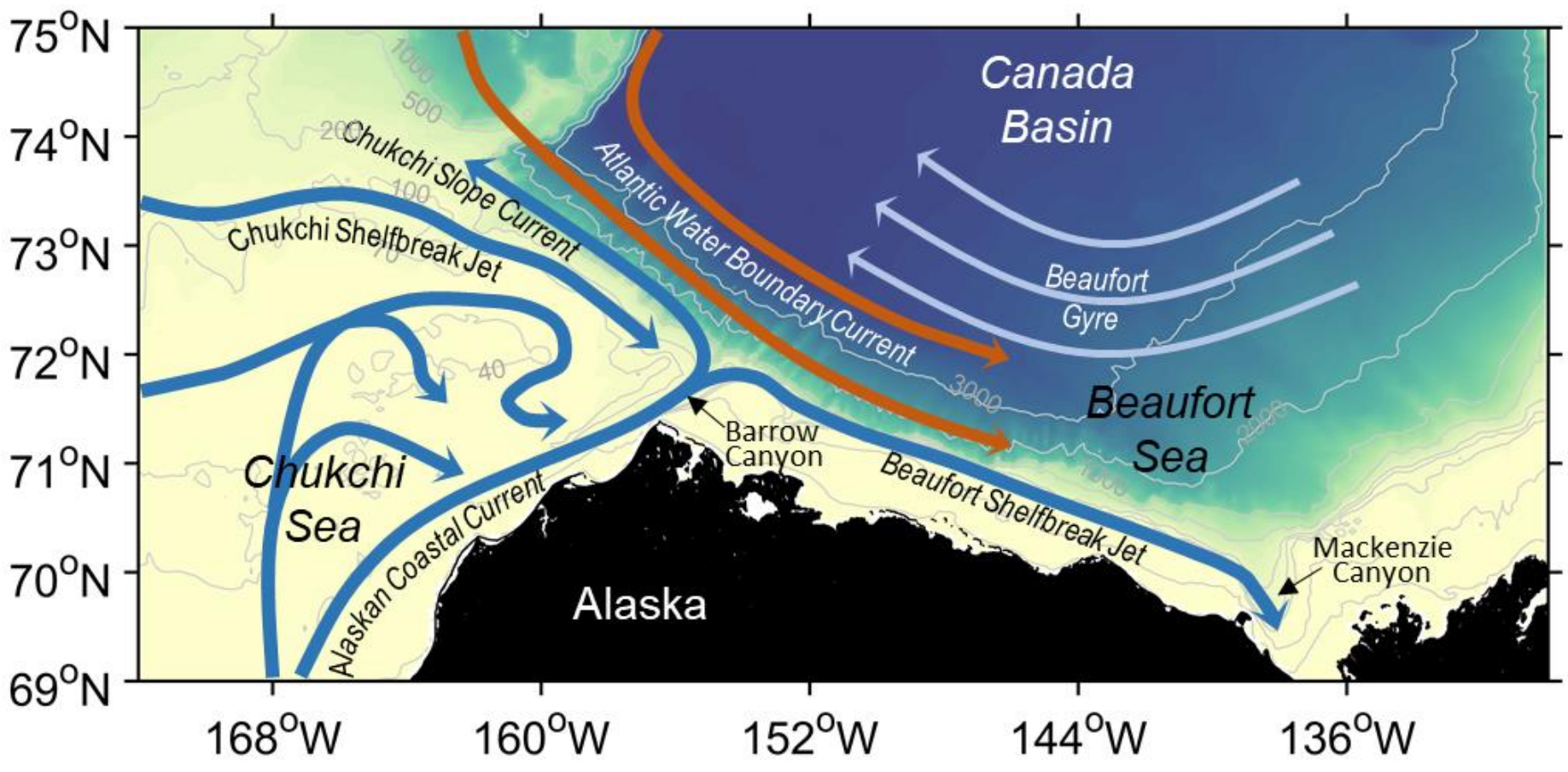

**Figure 1. Schematic circulation in the Chukchi and Beaufort Seas, from Leng et al. (2022).**

A year-long mooring array shows that the CSC is surface-intensified in summer/fall, but it becomes subsurface-intensified during winter/spring (Li et al., 2019). This is also supported by the results from a pan-Arctic numerical model (Leng et al., 2021). Using a simplified quasigeostrophic theoretical model, Leng et al. (2022) further demonstrated that the structure of the CSC is related to the ice condition. Typically, the slope current is laterally sheared and the corresponding spatially variable friction between the ice and the slope current can drive an overturning circulation to modify the thermal wind shear. This then results in the formation of a subsurface velocity maximum (via geostrophic setup) as seen in the mooring observations (Li et al., 2019).

It is well established that the "ice-ocean governor" - the negative feedback between the ice and ocean currents - acts to help equilibrate the BG via diminishing the wind-driven Ekman pumping (e.g., Meneghello et al., 2018, 2020; Dewey et al., 2018; Zhong et al., 2018; Doddridge et al., 2019). In addition, the ice friction can suppress surface instabilities (Meneghello et al., 2021) and dissipate existing surface eddies (Ou and Gordon, 1986), and this helps to explain the observed subsurface kinetic energy (KE) maxima in the interior Canada Basin. A recent idealized numerical study suggests that the dissipation of surface eddies relies much on the sea ice concentration (Shrestha and Manucharyan, 2022). When the sea ice concentration is high enough, the ice cover acts as a nearly immobile surface lid and thus dissipates the surface eddies effectively.

The Chukchi continental slope is one of the most energetic regions in the western Arctic Ocean as it is populated with strong boundary currents and mesoscale eddies (Kubryakov et al., 2021; Wang et al., 2020). These eddies, either formed via local baroclinic instability or propagated from other areas, are important in the exchanges of water masses and energy between the CSC and BG. During summer months, the CSC appears as a meandering free jet with opposing cross-stream gradients of potential vorticity (PV) within the current - a necessary condition for baroclinic instability (Corlett and Pickart, 2017).

However, it is not yet clear whether the CSC is baroclinically unstable during the ice-covered period and how the ice friction works on the energetics of the current and eddies.

In this paper, we investigate the slope current energetics using a set of experiments with an idealized primitive equation numerical model. Our focus is on the period when the slope region is covered by packed ice with a 100% concentration rather than the full melting-freezing cycle. Sect. 2 describes the model configuration and experimental design, and outlines the procedures for the energetics calculations and linear instability analysis. Sect. 3 shows results from the experiments and illuminates the role of ice friction in the slope current energetics. Finally, a summary and discussion are given in Sect. 4.

## 2 Methods

### 2.1 Model description

The Massachusetts Institute of Technology general circulation model (MITgcm) (Marshall et al., 1997) is used in this study. A meridional channel bounded by two solid walls is set up on a 300 km (width) by 200 km (length) Cartesian grid with an application of periodic boundary conditions at the northern and southern edges (see the schematic in Fig. 2). For discussion purposes axes $x$, $y$, and $z$ denote cross-channel, along-channel, and upward directions, respectively. The topography is idealized for a continental shelf ($x$ = 0 to 50 km, $z$ = –100 m), slope ($x$ = 50 to 90 km, slope = 0.01), and open ocean ($x$ = 90 to 300 km, $z$ = –500 m). The horizontal resolution is 1 km so that mesoscale eddies can be resolved. There are 48 levels in the vertical with the layer thickness increasing from 2 m at the surface layer to 25 m at the bottom layer. The Coriolis parameter is constant $f = 1.4 \times 10^{-4}\ \mathrm{s}^{-1}$.

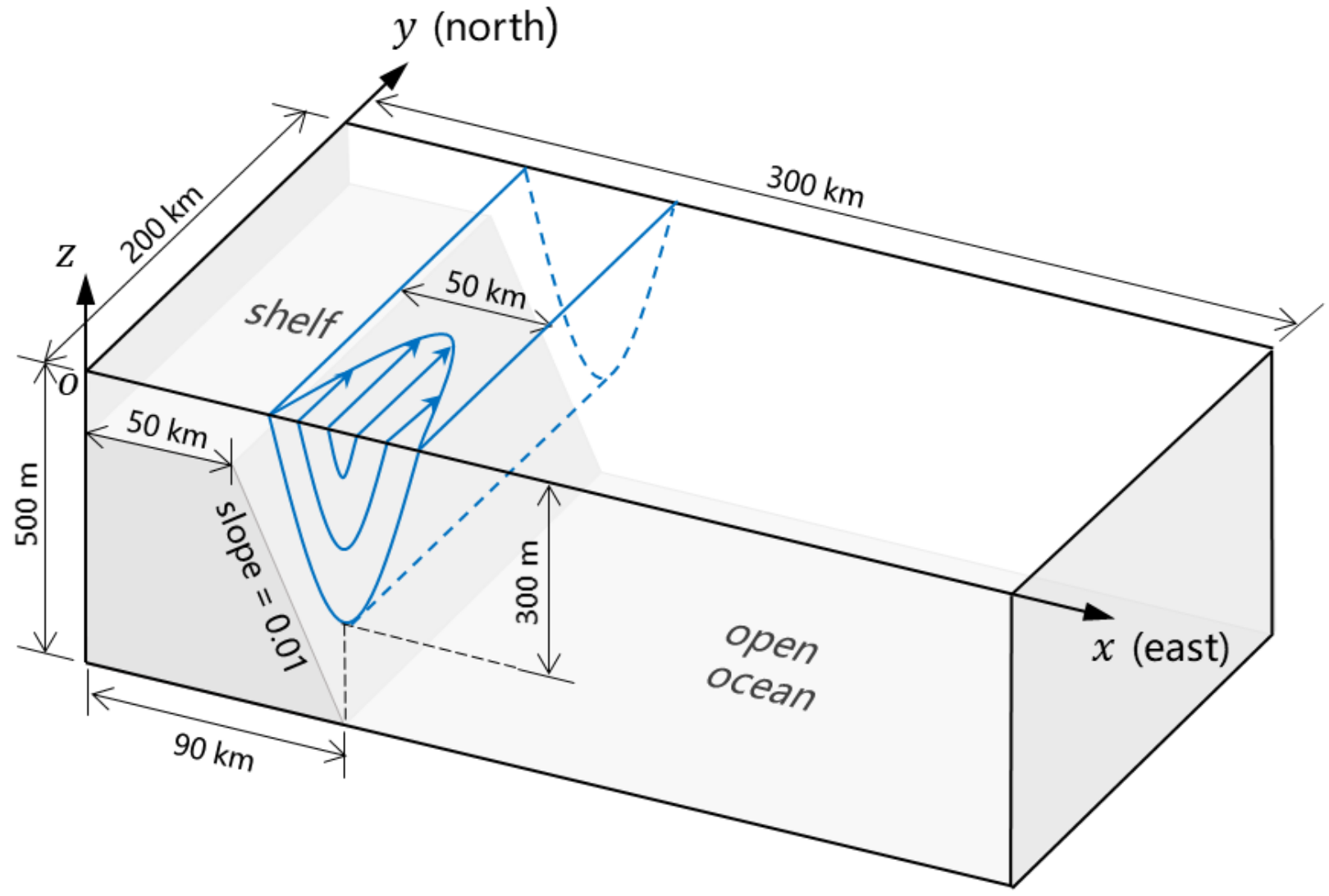

**Figure 2. Schematic of the model domain and bottom topography. The initial current flows northward along the channel with its onshore part overlying the continental slope.**

Parameterizations for this idealized model follow a pan-Arctic model that has been used to study the origin and fate of the CSC (Leng et al., 2021). The ocean model utilizes a nonlinear equation of state of seawater (Jackett and McDougall, 1995). Vertical mixing is calculated by the nonlocal K-Profile Parameterization (KPP) mixing scheme (Large et al., 1994). The KPP background diffusivity is small ($5.44 \times 10^{-7}$ m$^2$ s$^{-1}$) as required by the parameterization of salt plume (Nguyen et al., 2009). The modified horizontal viscosity scheme of Leith (1996) that can sense the flow divergence (Fox-Kemper and Menemenlis, 2008) is used with nondimensional Leith biharmonic viscosity factor 1.5. We also set a quadratic bottom drag with coefficient $2.0 \times 10^{-3}$ and free-slip lateral boundary conditions. The viscous-plastic dynamic-thermodynamic sea ice model of Zhang and Hibler (1997), as modified by Losch et al. (2010), is coupled to the ocean model.

## 2.2 Initial conditions

The model is initialized with a northward-flowing geostrophic current with its onshore part overlying the continental slope (Fig. 2). This is similar to the CSC flowing along the Chukchi slope (Corlett and Pickart, 2017). The initial along-channel velocity is given by (Fig. 3a)

$$v^{ini}(x,z) = \begin{cases} v_s^{ini}[\cos(\pi z/300) + 1]/2\,, & -300 \le z \le 0 \text{ m}, \\ 0, & -500 \le z < -300 \text{ m}, \end{cases} \quad (1a)$$

with the surface velocity

$$v_s^{ini}(x) = \begin{cases} V_m \sin[\pi(x-65)/L_0], & 65 \le x \le 115 \text{ km}, \\ 0, & 0 \le x < 65 \text{ km}, 115 < x \le 300 \text{ km}, \end{cases} \quad (1b)$$

where $V_m = 0.2$ m s$^{-1}$ is the maximum velocity and $L_0 = 50$ km is the flow width. The initial potential density field $\rho^{ini}$ (Fig. 3a, black contours) is derived from the velocity field with consideration of the thermal wind relation, $f_0(\partial v^{ini}/\partial z) = -(g/\rho_0)(\partial \rho^{ini}/\partial x)$, where $g = 9.8$ m s$^{-2}$ is the gravitational acceleration and $\rho_0 = 1028$ kg m$^{-3}$ is the reference density. The density profile at the flow center ($x = 90$ km) (Fig. 3c, red curve) is calculated from the December monthly mean salinity and temperature profiles at 72.625° N, 156.875° W (Fig. 3d), which are taken from the World Ocean Atlas climatology for the period 2005-17 (Zweng et al., 2018; Locarnini et al., 2018). The constructed density field shows a similar vertical structure as in the shipboard observations (see Fig. 4 in Corlett and Pickart (2017)). The initial salinity field $S^{ini}$ input to the model is derived from the density field by assuming $\nabla\rho^{ini} = \beta\nabla S^{ini}$, where $\nabla$ is the gradient operator and $\beta = 0.8$ kg m$^{-3}$ psu$^{-1}$ (note that temperature has little impact on density at near-freezing temperatures). We also input an initial temperature field $T^{ini}$ which satisfies $\nabla T^{ini} \times \nabla S^{ini} = 0$, i.e., the initial temperature is distributed following the isohalines (or isopycnals) (Fig. 3b). Using the constructed density field, we calculate the background mean stratification $N^2 = -(g/\rho_0)(\partial\rho_r/\partial z)$, where $\rho_r$ is the area-mean potential density as a function of depth. The resulting stratification reveals two peaks in the vertical, the shallower peak at a depth of –59 m and the deeper peak at –218 m (Fig. 3e). A similar double-peak feature is also shown in the stratification profiles computed from the hydrographic and mooring data in the interior Canada Basin (Meneghello et al., 2021). The model domain is covered by sea ice throughout the simulation (the ice concentration $\alpha = 1$). The initial ice thickness is 1 m at each model grid. For maintaining the sea ice, we give a constant downward longwave radiation (180 W m$^{-2}$) and

shortwave radiation (20 W m$^{-2}$) as in Leng et al. (2022). There is no wind forcing for the simulation and a minimum surface boundary layer thickness of 10 m is prescribed.

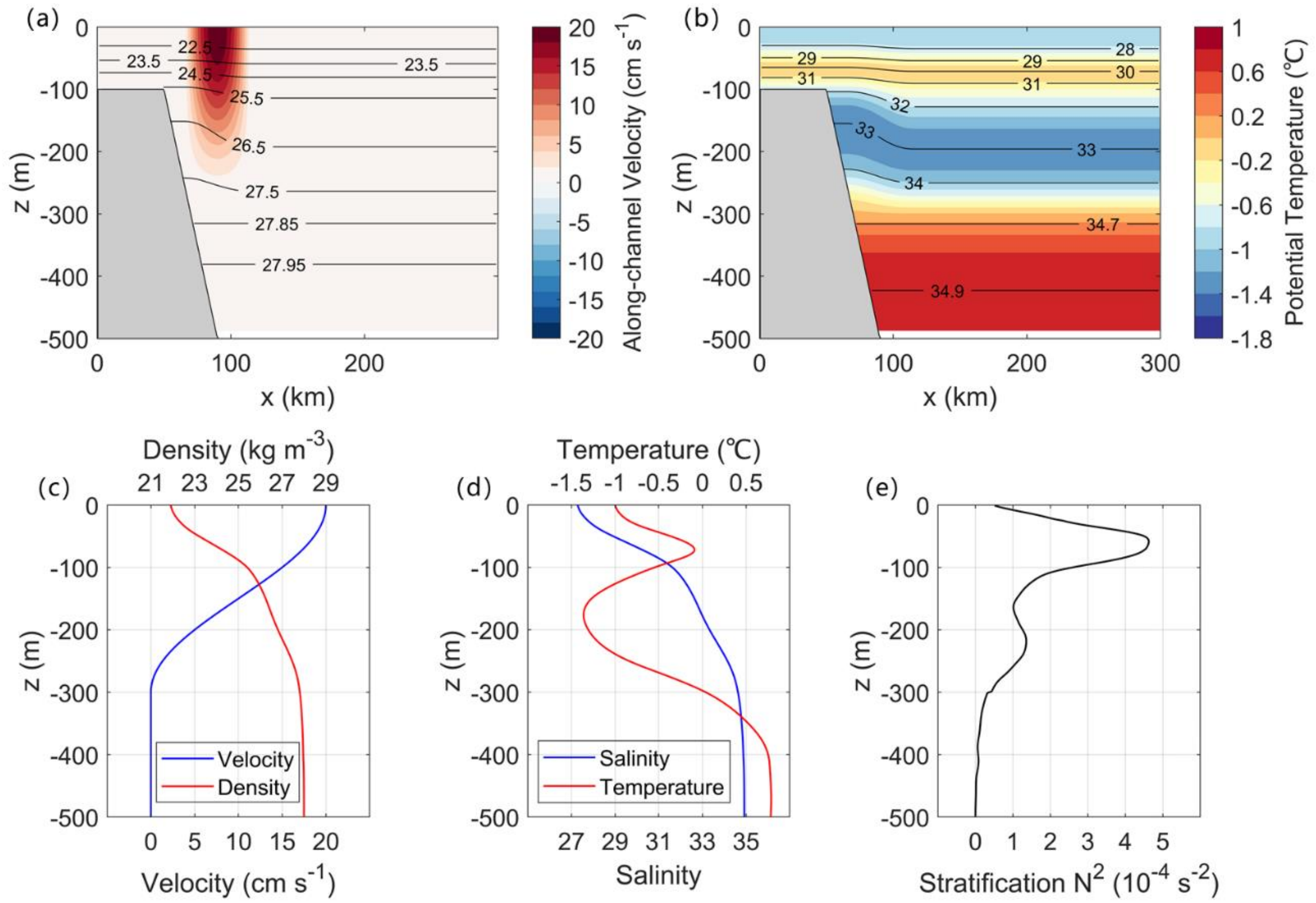

**Figure 3. Initial conditions for the numerical experiments: (a) along-channel velocity (color) and isopycnals (black contours), (b) potential temperature (color) and isohalines (black contours), (c) vertical profiles of the along-channel velocity (blue) and density (red) at the flow center (*x* = 90 km), (d) vertical profiles of the salinity (blue) and temperature (red) at *x* = 90 km, and (e) vertical profile of the background mean stratification $N^2$.**

### 2.3 Experimental design

The ice-ocean stress in the model is calculated from the ice velocity $\mathbf{u}_i$ and sea surface velocity $\mathbf{u}_s$, following a quadratic drag law (Zhang and Hibler, 1997; Losch et al., 2010),

$$\boldsymbol{\tau} = \alpha \rho_0 C_{Di} |\mathbf{u}_i - \mathbf{u}_s| (\mathbf{u}_i - \mathbf{u}_s), \quad (2)$$

where $C_{Di} = 5.5 \times 10^{-3}$ is the drag coefficient. Five experiments are carried out to diagnose the effects of ice friction, including a base experiment (Exp-PD) and four sensitivity experiments (Exp-D, Exp-P, Exp-PD100, and Exp-PF; P, D, and F stand for initial perturbation, ice-ocean drag, and forcing, respectively) (Table 1). In Exp-PD, the ice-ocean drag is turned on and a random perturbation ($\tilde{u}$, $\tilde{v}$) is added to the initial velocity field to produce instabilities. The perturbation is weighted by $0.05v^{ini}$ and given as

$$\begin{cases} \tilde{u} = 0.05 v^{ini} A_1, \\ \tilde{v} = 0.05 v^{ini} A_2, \end{cases} \quad (3)$$

where $A_1$ and $A_2$ are three-dimensional random scalar fields drawn from the standard normal distribution (white noise). The resulting perturbation is of order 1 cm $s^{-1}$.

In Exp-D, we keep the ice-ocean drag and remove the initial perturbation to focus on the evolution induced by the ice friction (note that this is nearly a two-dimensional calculation). In Exp-P, we turn off the ice-ocean drag and include the initial perturbation. This allows instabilities to grow without the influence of surface forces. Exp-PD100 is a restart of Exp-P from day 100 and is run for 100 days with the ice-ocean drag (the initial conditions for Exp-PD100 are taken from the output of Exp-P on day 100). This restart experiment aims to examine the effect of ice friction on pre-existing eddies. The results for the first 100 days of Exp-PD100 presented below are actually from Exp-P. The last experiment, Exp-PF, has no seaice model and is forced by the surface stress, shortwave radiation, and heat and freshwater fluxes at the surface of the ocean diagnosed from the daily output of Exp-PD. This will help distinguish the relative influence of baroclinic instability versus Ekman pumping on the release of APE. Each simulation is integrated for 200 days except for Exp-PD100.

**Table 1. The setup for the base and sensitivity experiments.**

| Experiment | Initial perturbation | Ice-ocean drag | Forcing | Period (days) |
|---|---|---|---|---|
| Exp-PD | Yes | Yes | No | 200 |
| Exp-D | No | Yes | No | 200 |
| Exp-P | Yes | No | No | 200 |
| Exp-PD100 | Yes | Yes | No | 100 |
| Exp-PF | Yes | No | Yes | 200 |

### 2.4 Energetics calculations

We calculate the KE and available potential energy (APE) per unit volume (in units of J $m^{-3}$) by

$$E_k = \frac{1}{2}\rho(u^2 + v^2), \tag{4}$$

and

$$E_p = \frac{g^2(\rho - \rho_r)^2}{2\rho_0 N^2}, \tag{5}$$

with $\rho$ being the output varying density and $u$, $v$ the cross- and along-channel velocity components. Note that the vertical velocity $w$ (order $10^{-5}$ m $s^{-1}$) is much smaller than $u$ and $v$ (order 0.1 m $s^{-1}$) and thus has little contribution to KE. It may be not appropriate to separate the velocity field into its time-mean $(\bar{u}, \bar{v})$ and time-varying $(u', v')$ parts and define the eddy kinetic energy (EKE) by $\rho(u'^2 + v'^2)/2$. This is because the slope current would change with time as a response to the ice friction such that $v'$ is nonzero even not encountering eddies. Namely, the EKE may be overestimated by $\rho(u'^2 + v'^2)/2$. Here we define the KE of the along-slope mean flow by

$$\widehat{E_k} = \frac{1}{2}\rho\hat{v}^2, \tag{6}$$

where $\hat{v}$ is the along-channel average of $v$. Then, we use the difference $E_k - \widehat{E_k}$ to measure the EKE.

The rate at which the surface stress works on the ocean, i.e., the power input to the ocean from surface stress (in units of W $\mathrm{m}^{-2}$), can be written as a product of surface stress and surface current velocity (e.g., Wunsch, 1998; Zhai et al., 2012; Zhong et al., 2019). For the whole model domain, the power of ice friction (in units of W) is estimated by

$$P_\tau = \int_0^X \int_0^Y (\boldsymbol{\tau} \cdot \mathbf{u}_s)\, dxdy, \tag{7}$$

where $X$ = 300 km and $Y$ = 200 km. The time integral of $P_\tau$ gives the work done by the ice friction

$$W_\tau(t) = \int_0^t P_\tau\, dt. \tag{8}$$

### 2.5 Linear instability calculations

We analyze the baroclinic instability of the slope current based on the linearized quasigeostrophic PV equation (Smith, 2007; Tulloch et al., 2011; Meneghello et al., 2021)

$$\left(\frac{\partial}{\partial t} + \mathbf{U}\cdot\nabla_h\right) q + \mathbf{u}\cdot\nabla_h Q = 0, \tag{9a}$$

$$q = \nabla_h^2\psi + \frac{\partial}{\partial z}\left(\frac{f^2}{N^2}\frac{\partial\psi}{\partial z}\right), \tag{9b}$$

$$\nabla_h Q = -\frac{\partial}{\partial z}\left(\frac{f^2}{N^2}\mathbf{k}\times\frac{\partial\mathbf{U}}{\partial z}\right), \tag{9c}$$

where $q$ is the perturbation PV, $\psi$ is the perturbation streamfunction, and $\mathbf{u} = \mathbf{k}\times\nabla_h\psi$ is the perturbation horizontal velocity ($\mathbf{k}$ is the upward unit vector and $\nabla_h$ is the horizontal gradient operator). The background PV gradient $\nabla_h Q$ on the $f$-plane has been simplified as a function of the background horizontal velocity $\mathbf{U}$ and stratification $N^2$ (note that the relative vorticity of the mean flow is neglected and $\nabla_h Q$ varies only with depth). Surface ($z = 0$) and bottom ($z = -h$) boundary conditions for Eq. (9) are provided by (Williams and Robinsonm, 1974; Pedlosky, 1987; Meneghello et al., 2021)

$$\left(\frac{\partial}{\partial t} + \mathbf{U}\cdot\nabla_h\right)\frac{\partial\psi}{\partial z} = \frac{\partial\mathbf{U}}{\partial z}\cdot\nabla_h\psi - \frac{N^2}{f}\frac{d_s}{2}\nabla_h^2\psi, \quad z = 0, \tag{10a}$$

$$\left(\frac{\partial}{\partial t} + \mathbf{U}\cdot\nabla_h\right)\frac{\partial\psi}{\partial z} = \frac{\partial\mathbf{U}}{\partial z}\cdot\nabla_h\psi + \frac{N^2}{f}\frac{d_b}{2}\nabla_h^2\psi, \quad z = -h, \tag{10b}$$

where $d_s$ and $d_b$ are the surface and bottom Ekman layer depths, and $d_s\nabla_h^2\psi/2$ and $d_b\nabla_h^2\psi/2$ the corresponding Ekman pumping. Typically, the surface Ekman layer depth of order 10 m is equivalent to the vertical diffusivity $\upsilon_E = d_s^2 f/2$ of approximately $10^{-2}$ $\mathrm{m}^{-2}$ $\mathrm{s}^{-1}$.

We assume a plane-wave solution of the form $\psi = \hat{\psi}(z)e^{i(ly-\omega t)}$, where $\hat{\psi}$ is the vertical structure of the perturbation, $l$ is the along-channel wavenumber, and $\omega = \omega_r + i\omega_i$ is the complex frequency. If the imaginary frequency component $\omega_i$ is

positive, the corresponding mode is unstable and will grow exponentially with time on a time scale of $1/\omega_i$. Substitution of the plane-wave solution into Eq. (9) yields a generalized eigenvalue problem, with $\omega$ being the eigenvalues and $\hat{\psi}$ the eigenvectors. The eigenvalue problem is then discretized on the staggered vertical model grid and solved numerically following the algorithm proposed by Smith (2007). Bottom topography is not considered as we calculate the background PV gradient from the velocity and density profiles at the center of the front, offshore of the continental slope.

## 3 Results

### 3.1 Base experiment

#### 3.1.1 Generation of subsurface eddies

In the initial state the slope current is surface-intensified, so the normalized relative vorticity $\zeta/f$ is larger at the surface than the subsurface ($z$ = –105 m) (Figs. 4a and b). Due to the presence of ice friction in Exp-PD, the surface vorticity decays quickly with time while the subsurface vorticity remains large (Figs. 4c-h). On day 60 the perturbations are visible in the subsurface (Fig. 4d), which then grow into eddies as shown in Figs. 4f and h. The growth of subsurface eddies corresponds to an increase in total KE (Fig. 5a, black curve), meanwhile, the mean flow KE remains small and starts to decrease on day 100 (Fig. 5a, red curve). This gives rise to a continuous increase in EKE since day 60 (the distance between the two curves keeps increasing).

185

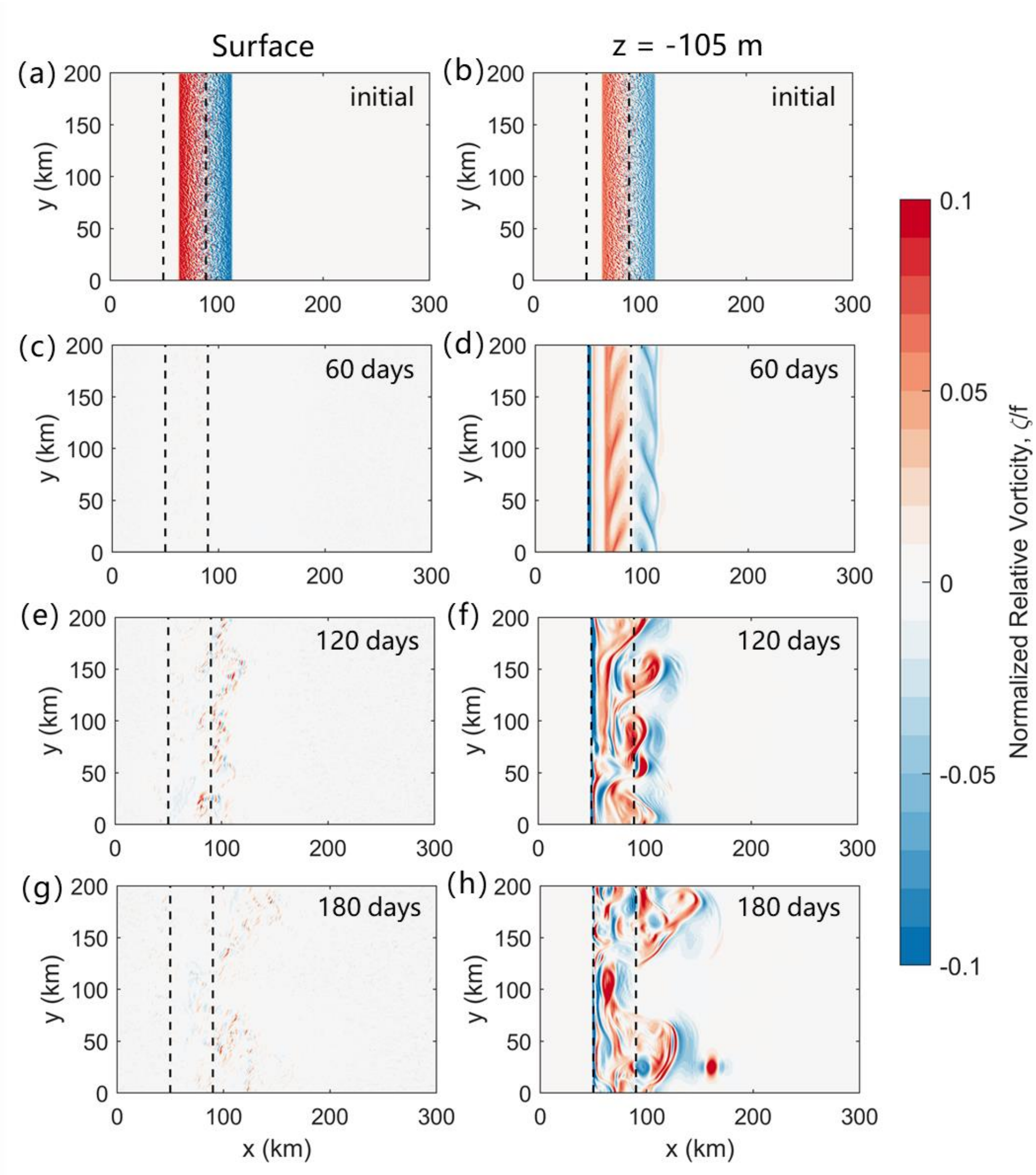

**Figure 4. Distributions of the relative vorticity $\zeta = \partial v/\partial x - \partial u/\partial y$ normalized by $f$ at $t$ = 0, 60, 120, and 180 days from Exp-PD: (left) at the surface and (right) at the depth of –105 m. The region of the continental slope is outlined by two black dashed lines in each panel.**

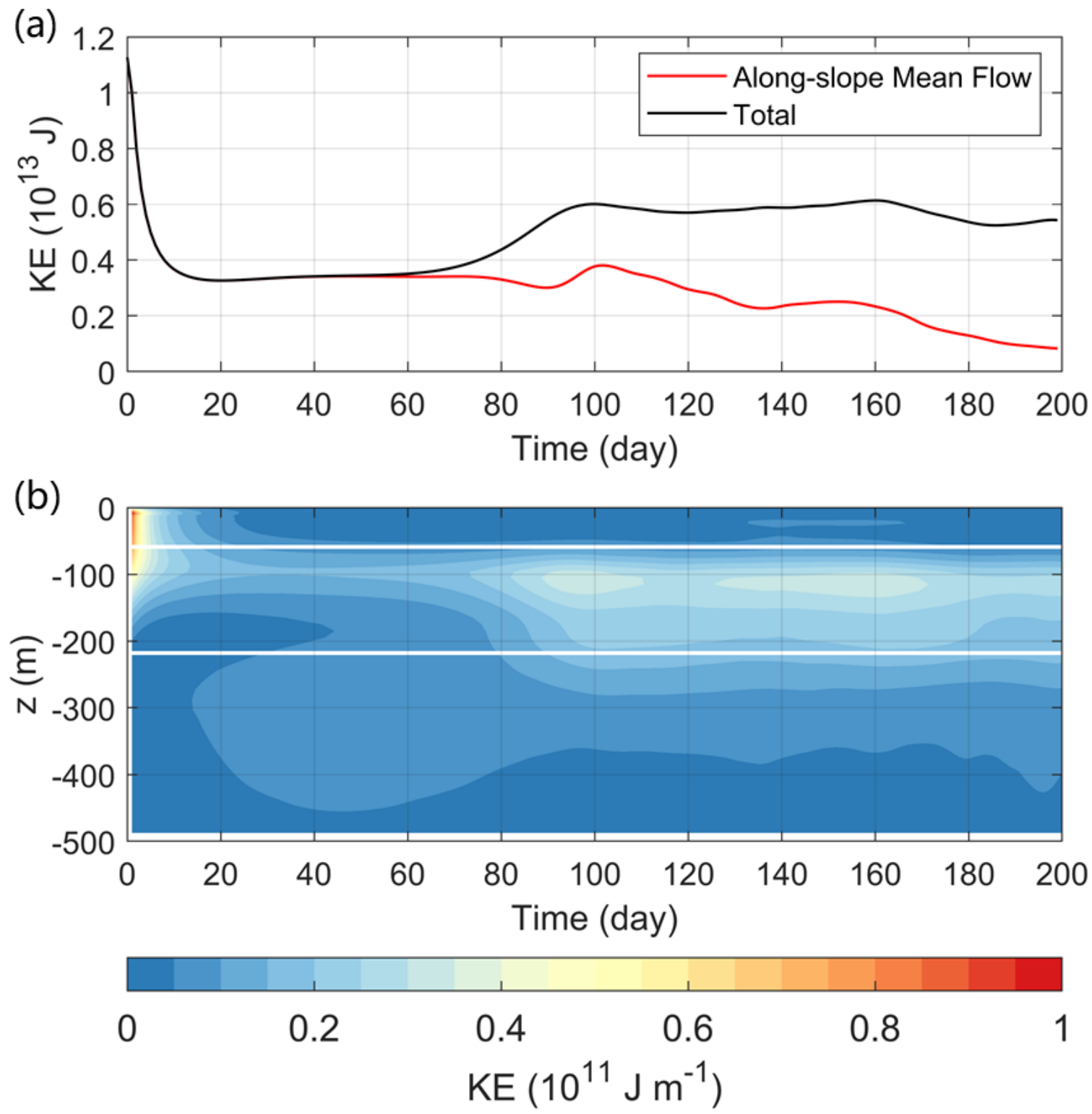

**Figure 5. (a) Time evolution of the KE of the along-slope mean flow (red curve) and the total KE integrated over the model domain (black curve) from Exp-PD. (b) Time-depth plot of the horizontal integral of KE from Exp-PD. The white lines mark the depths of the two stratification peaks (–59 and –218 m).**

The mechanism for the generation of subsurface eddies in interior Arctic Ocean has been investigated by Meneghello et al. (2021) through linear baroclinic instability analysis. Specifically, they found that the strong stratification at the depth of around –50 m can protect the underlying unstable modes from the effect of ice friction. With the output density and velocity profiles at $x$ = 90 km on day 60, we calculate the growth rate $\omega_i$ and amplitude $|\hat{\psi}|$ for the wavenumber range $10^{-6} < l < 1\ \mathrm{m}^{-1}$. The results show three unstable branches, including a halocline branch and two surface branches (Fig. 6).

The surface unstable modes have length scales (defined as $1/l$) ranging from a few meters to 10 km so that only the longest surface waves can be fully resolved by the numerical model. Note that a few small-scale signals are visible in the surface vorticity field (e.g., Figs. 4e and g). The fastest-growing mode for the halocline branch has a length scale of 10 km, and its amplitude reaches the maximum at a depth of around –100 m, in line with the subsurface EKE maximum in Exp-PD (Fig. 6b). Therefore, the generation of subsurface eddies should be due to the growth of the halocline mode. It is noted that the halocline mode grows at a time scale of ~6 days, much faster than the growth of the total KE (time scale of one month). This is because the background velocity and density profiles are taken from the center of the front, where the PV gradient is the strongest. In other regions the PV gradient is weaker and the corresponding growth rate is smaller. It takes time for the eddies to spread

over the model domain. The linear instability calculations may also overestimate the growth rate since the interior viscosity is not considered.

The ice friction is known to suppress the growth of surface unstable modes (Meneghello et al., 2021). As we change the surface Ekman layer depth from 0 to 20 m, the maximum growth rate of the surface modes decreases from ~2.2 to 0.4 per day (Fig. 6a, blue lines). However, the growth of the halocline mode is not sensitive to the ice friction (Fig. 6a, black lines), and varying the surface Ekman layer depth only cause a slight change in the vertical structure (Fig. 6b, black lines).

Although the ice friction has little impact on the linear instability of the halocline mode, it can also affect the subsurface EKE through Ekman pumping. When the sea surface relative vorticity is nonzero, the spatially variable friction between the ice and surface flows would drive Ekman pumping to decrease APE and EKE over a vertical scale $H = fL/N$, where $L$ is the horizontal length scale of the flow. If we take $f = 10^{-4}\ \mathrm{s}^{-1}$, $L = 10$ km, and $N = 10^{-2}\ \mathrm{s}^{-1}$, the vertical scale is estimated to be 100 m (note that Ekman pumping could affect deeper depths for larger scale flows). Note that this is much deeper than the Ekman layer, where the direct influence of surface stresses is confined. Fig. 6b shows that the deep EKE in Exp-PD approximately follows the eigenvalue profile but the EKE above –100 m decays more quickly towards the surface. This demonstrates the Ekman spin-down influence of sea ice on EKE and is not included in the linear stability analysis.

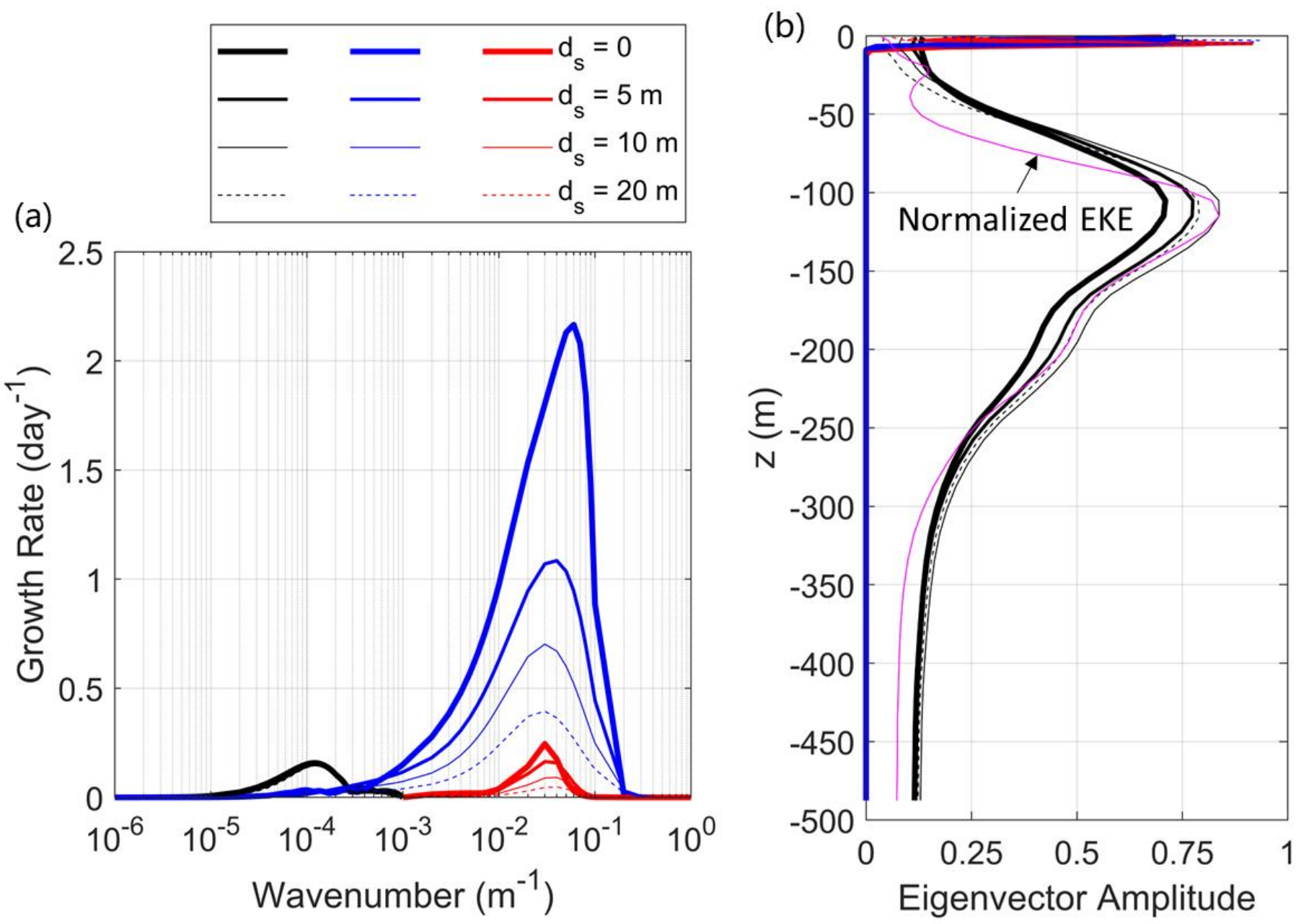

**Figure 6. (a) Growth rate $\omega_i$ of unstable modes as a function of wavenumber for the surface Ekman layer depth $d_s$ = 0, 5, 10, and 20 m, and bottom Ekman layer depth $d_b = 20$ m. Three unstable branches are resolved, including a halocline branch (black) and two surface branches (blue and red). (b) Eigenvector amplitude $|\hat{\psi}|$ of the most unstable mode for each unstable branch. The magenta curve in (b) is the vertical profile of EKE (from Exp-PD, averaged over final 100 days) normalized by the maximum eigenvector amplitude of the halocline mode.**

### 3.1.2 Evolution of the flow structure

Figure 7a shows a symmetric distribution of downwelling and upwelling at $t$ = 5 days, which is driven by the spatially variable friction between the ice and surface mean flow (Leng et al., 2022). The streamfunction obtained by integrating the vertical velocity along $x$ presents an anti-clockwise overturning circulation (Fig. 7b). Such overturning acts to modify the thermal wind shear via moving the isopycnals, and this then results in the formation of the subsurface velocity core at a depth of approximately –100 m and the counter current extending from the shelfbreak to the open ocean (via geostrophic setup) (Fig. 7c). The onshore part of the counter current is trapped by the sloping bottom, with the maximum velocity occurring in the vicinity of the shelfbreak. Offshore of the continental slope, the counter current is weaker and spans wider ranges of distance and depth.

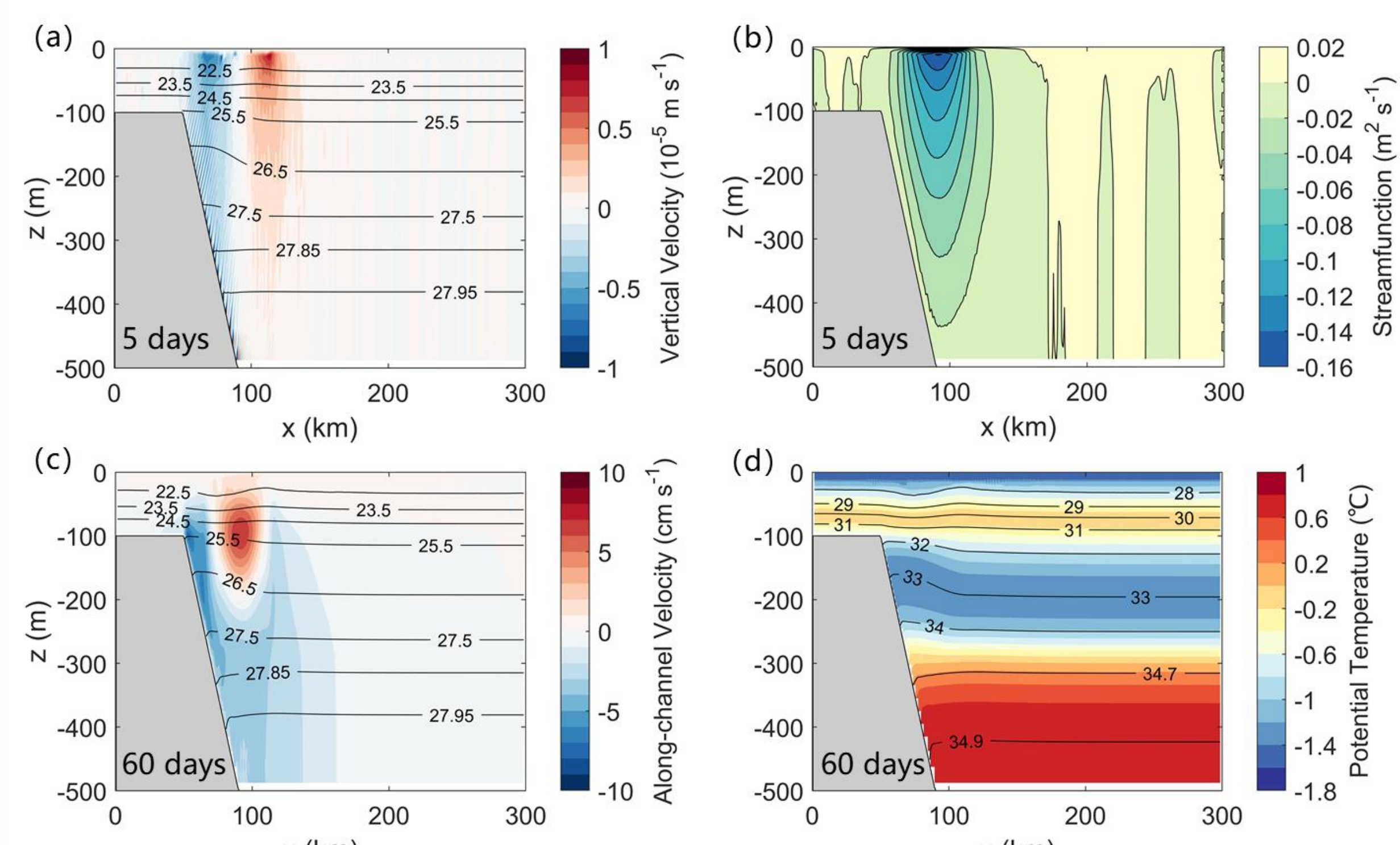

**Figure 7. (a) Vertical velocity (color) overlain by isopycnals (black contours) and (b) overturning streamfunction averaged between $y$ = 0 and 200 km at $t$ = 5 days. (c) Along-channel velocity (color) overlain by isopycnals (black contours) and (d) potential temperature (color) overlain by isohalines (black contours) averaged between $y$ = 0 and 200 km at $t$ = 60 days.**

One may think that the ice-induced overturning would produce diapycnal transport since the overturning streamlines intersect the isopycnals. However, the isopycnals get displaced instead of staying at the same depth, and meanwhile, the overturning diminishes quickly along with the decay of Ekman pumping. In the steady state (before the generation of eddies) the surface velocity and overturning are very weak and there are no vertical motions contributing to diapycnal transport. This is why the potential temperature remains distributed following isohalines (or isopycnals) at $t$ = 60 days (Fig. 7d).

### 3.2 Comparison between the base and sensitivity experiments

#### 3.2.1 Evolution of KE and APE

As mentioned above in Sect. 3.1, the time evolution of KE in Exp-PD should be related to the ice-induced Ekman pumping as well as the generation of subsurface eddies through baroclinic instability. This can be further illustrated by the results from the sensitivity experiments. In Exp-D, the initial perturbation is removed and a rapid loss of KE occurs in the first 10 days as in Exp-PD (Fig. 8a, black solid and dashed curves). Since no eddies are generated in Exp-D, the KE remains low until the end of the simulation. The ice-ocean drag is turned off in Exp-P, hence there is almost no change in KE in the first 60 days (Fig. 8a, blue solid curve). The increase of KE from day 60 to 100 is $\sim 1 \times 10^{13}$ J in Exp-P, three times that in Exp-PD. Figure 9b shows two KE maxima in the vertical in Exp-P after day 80. The deeper KE maximum at a depth of around –100 m is due to the growth of the halocline mode, while the upper KE maximum at the surface arises from both the surface and halocline modes (the halocline mode extends to the surface, see Figs. 10c and d). In the restart experiment (Exp-PD100), the KE drops quickly since day 100 as a response to the ice friction (Fig. 8a, blue dashed curve). The reduction of KE in Exp-PD100 mainly occurs over the upper 60 m, while the subsurface KE shielded by the shallower stratification peak is maintained to the end of the calculation (Fig. 9c).

Exp-PF includes the same Ekman pumping as Exp-PD but there is no damping of eddies by interaction with ice. Therefore, Exp-PF produces higher EKE than Exp-PD but lower EKE than Exp-P after day 60 (Fig. 8a). This demonstrates that the surface stress suppresses EKE in two ways. Exp-PF contains the large-scale Ekman pumping signal that reduces APE, and thus the source for EKE, and thus has lower EKE than Exp-P. However, it does not damp the perturbations directly so EKE is higher than for Exp-PD, mostly in the upper 50 m.

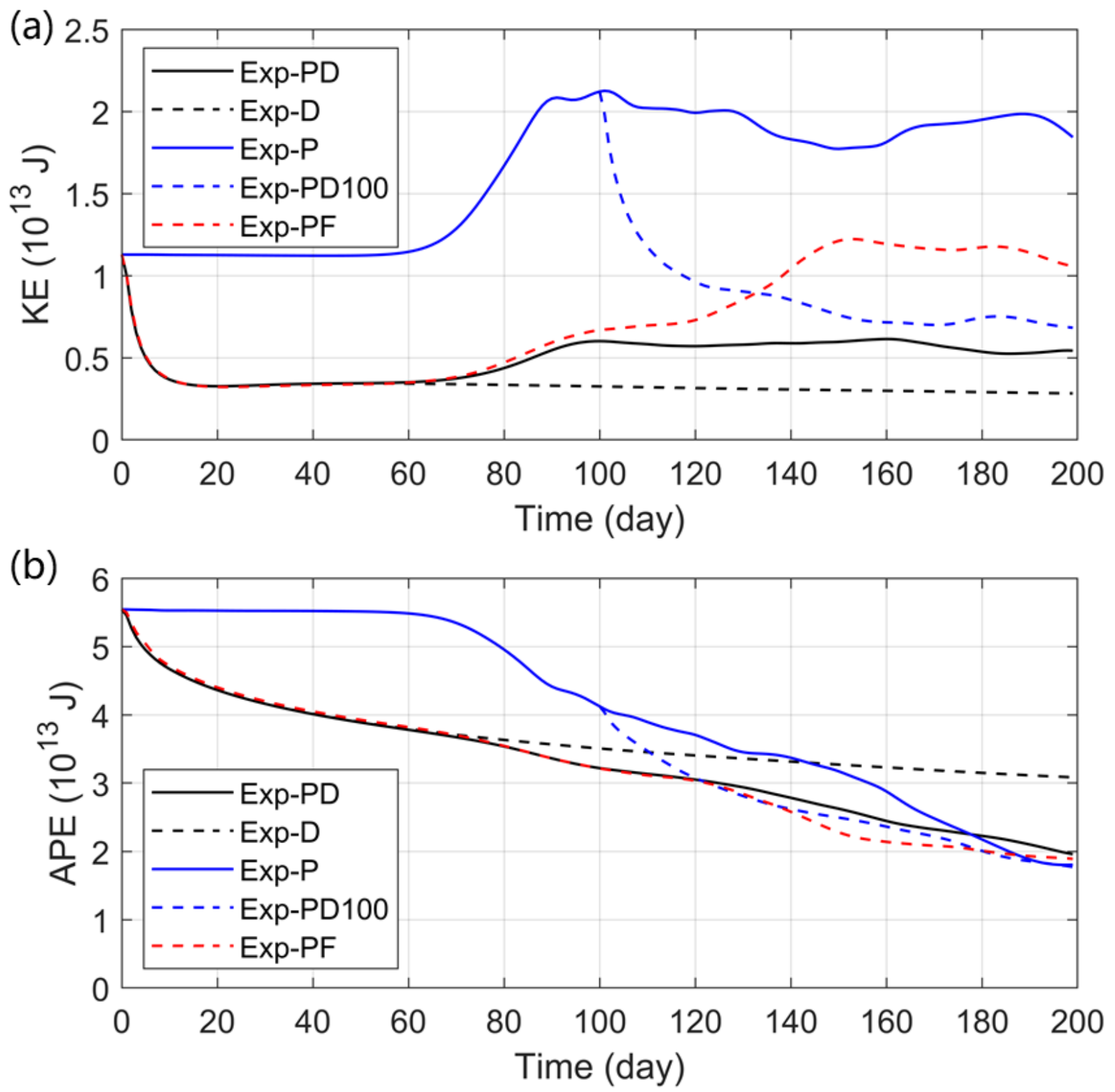

**Figure 8. Time evolution of the (a) KE and (b) APE integrated over the model domain from the base and sensitivity experiments.**

The evolution of APE is also sensitive to the ice friction (Fig. 8b). In the absence of ice-ocean drag (Exp-P), the APE appears to be steady in the first 60 days and then it undergoes a rapid reduction ($\sim 1.5 \times 10^{13}$ J) between day 60 and 100 (Fig. 8b, blue solid curve), as the KE increases by $\sim 1 \times 10^{13}$ J due to the growth of eddies (Fig. 8a, blue solid curve). Namely, the eddies draw energy from the APE stored in the slope current via baroclinic conversion. This differs from Exp-D where the total mechanical energy (both KE and APE) decreases since the beginning (Figs. 8a and b, black dashed curve) and the baroclinic conversion is not allowed because the flow is two-dimensional. Such loss in mechanical energy is due to the negative work done by the ice friction (this is discussed below in Sect. 3.2.2). Exp-PF has APE similar to Exp-PD but higher EKE because it does not lose EKE of the eddies to surface stress.

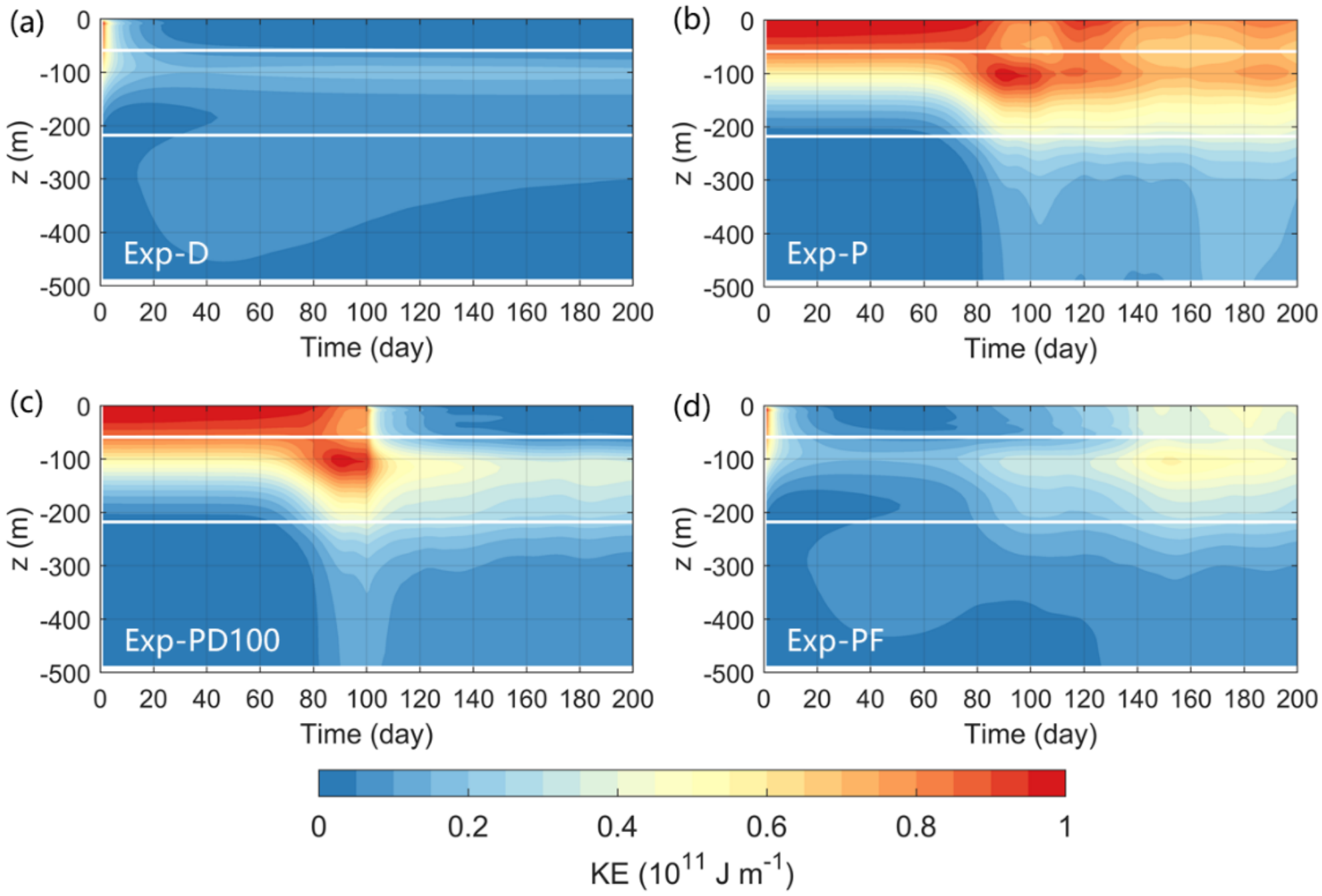

275 **Figure 9. Time-depth plot of the horizontal integral of KE from the four sensitivity experiments. The white lines in each panel mark the depths of the two stratification peaks (–59 and –218 m).**

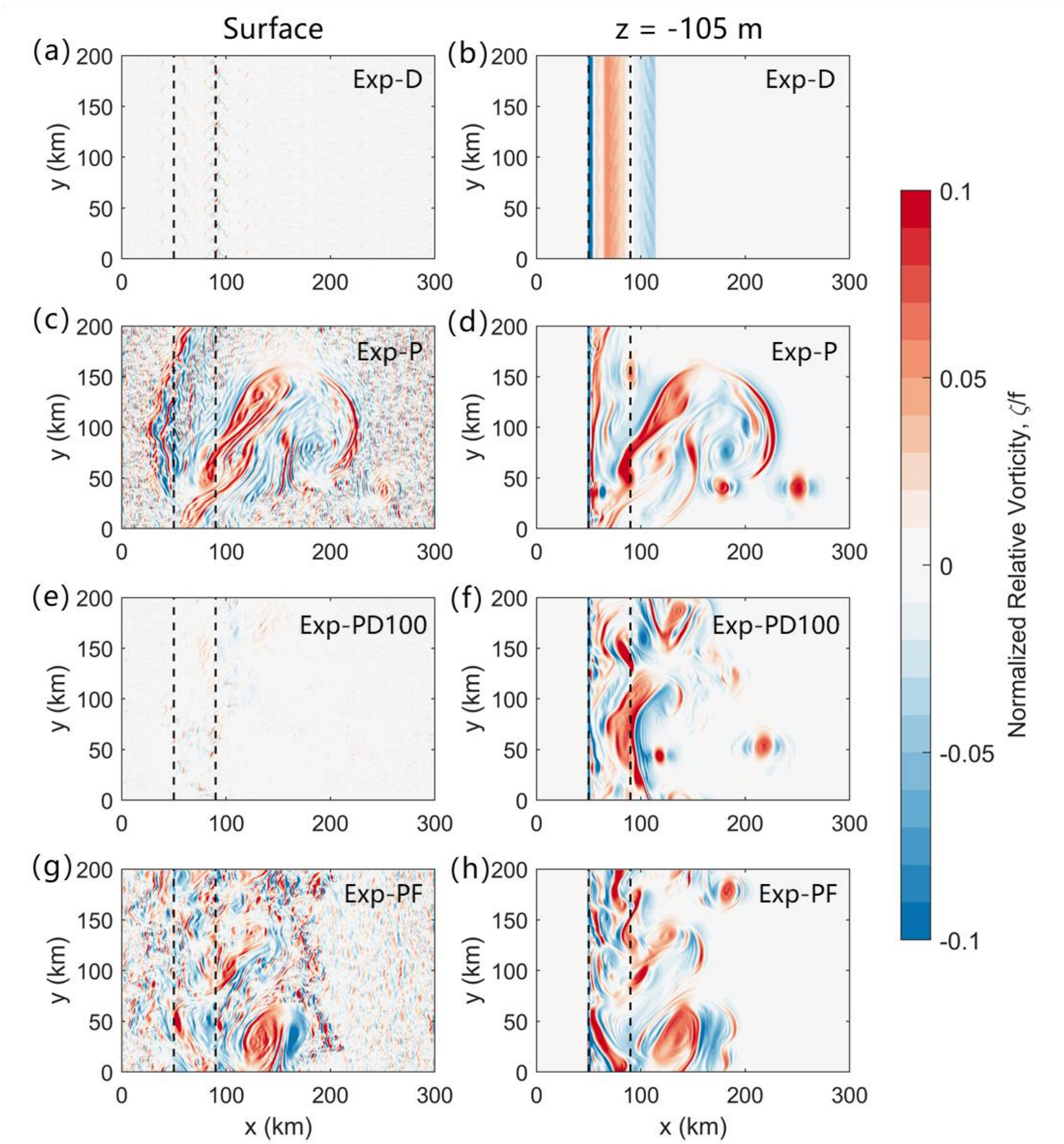

**Figure 10. Distributions of the normalized relative vorticity $\zeta/f$ from the four sensitivity experiments at *t* = 180 days: (left) at the surface and (right) at the depth of –105 m. The region of the continental slope is outlined by two black dashed lines in each panel.**

### 3.2.2 Work done by the ice friction

280 The ice cover acts as a nearly immobile surface lid in our experiments ($|\mathbf{u}_i| < 10^{-4}$ m s$^{-1}$) because the ice concentration maintains high and the ice internal stress is strong enough to counteract the ice-ocean stress and prevent the ice from drifting together with the current. Therefore, the ice-ocean stress is nearly in the opposite direction to the sea surface velocity ($\boldsymbol{\tau} \cdot \mathbf{u}_s < 0$) and does negative work on the ocean. In both Exp-PD and Exp-D the work done by the ice friction has reached $-2 \times 10^{13}$ J

by the end of the simulation (Figs. 11a and b, red curves), contributing to more than half of the loss in mechanical energy (Figs. 11a and b, black curves). The rest of the energy loss should be due to the interior and bottom frictions (measured by the difference between the black and red curves), which increases with time and appears to be more significant in the cases where eddies are well developed (Figs. 11c and d). Since the ice-ocean drag is turned off in Exp-P, there is no mechanical energy lost through the ice-ocean interface (Fig. 11c). In this case, the mechanical energy can be dissipated only by interior and bottom frictions. In Exp-PD100 the ice friction only works for 100 days, while it finally results in a reduction of $-2 \times 10^{13}$ J in mechanical energy, comparable to the dissipation due to interior and bottom frictions (Fig. 11d).

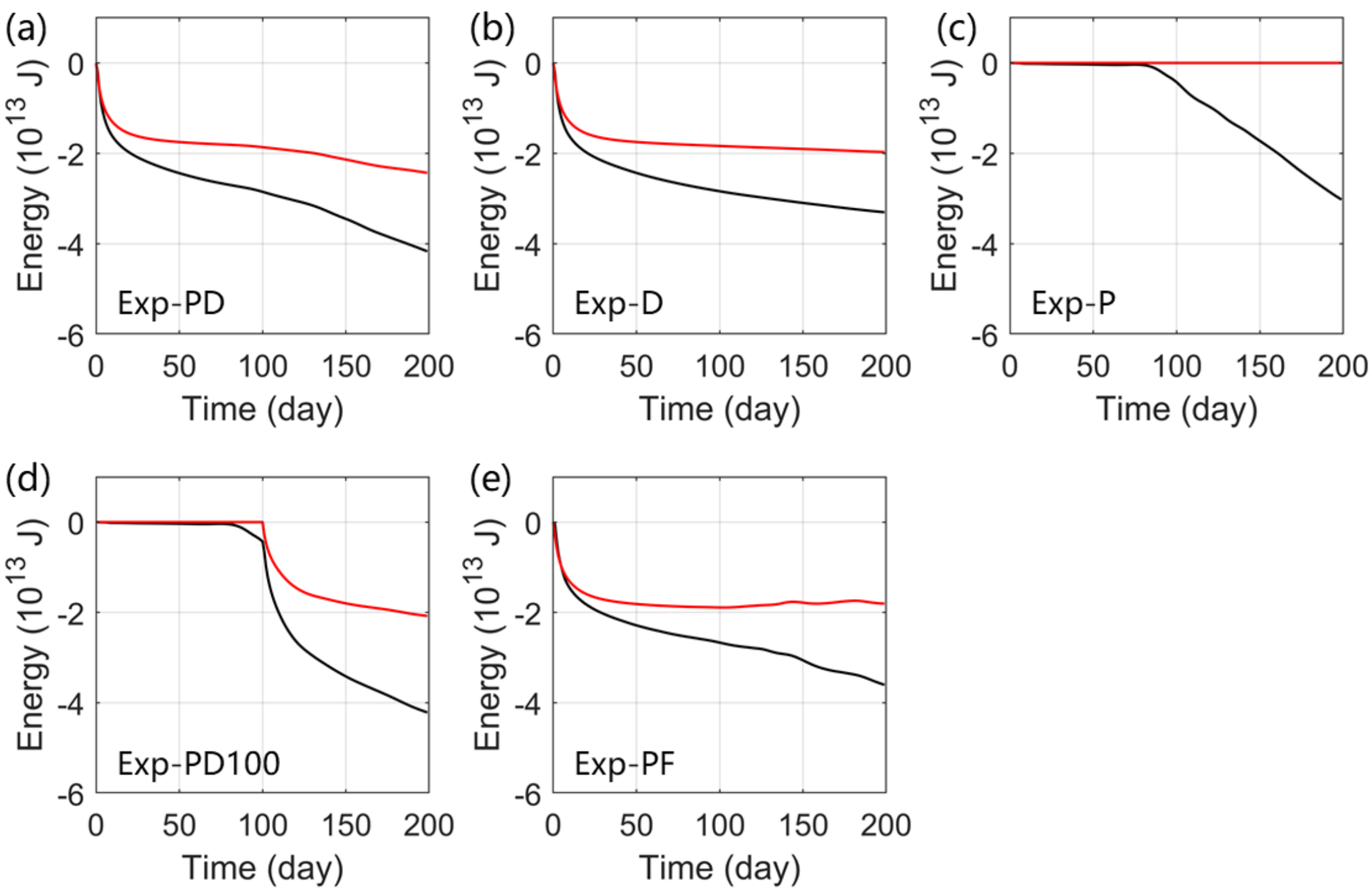

**Figure 11. The change of mechanical energy relative to *t* = 0 (black curve) and the work done by the surface stress (red curve) integrated over the model domain as functions of time from the base and sensitivity experiments.**

As mentioned above in Sect. 3.1.2, the ice friction drives Ekman pumping to modulate the isopycnal slopes, and this is likely to cause the release of APE. To better illustrate the role of ice-induced Ekman pumping, we introduce a simplified density equation

$$\frac{\partial \rho}{\partial t} + w \frac{\partial \rho_r}{\partial z} = 0, \tag{11}$$

which means that the local density would change as the vertical velocity advects a background mean density gradient (the horizontal advection terms are neglected). This is also the quasigeostrophic approximation and is valid for the large-scale

motions with Rossby number being much less than one. Combining Eqs. (5) and (11), we obtain a rate at which the APE changes (in units W m$^{-3}$)

$$\frac{\partial E_p}{\partial t} = g(\rho - \rho_r)w, \tag{12}$$

where $g(\rho - \rho_r)w$ is the vertical buoyancy flux. This equation suggests that the APE would be released if there is an upwelling advecting lighter water (relative to the background mean density) upward or a downwelling advecting denser water downward. This is the case in Exp-D where the ice-induced Ekman pumping (see e.g., Fig. 7a) results in a negative buoyancy flux that dominates the release of APE (Fig. 12b, the two curves are nearly identical). Furthermore, the integrated vertical buoyancy flux generally agrees with the rate of change of APE in the other four experiments (Figs. 12a, c-e), suggesting that Eq. (12) is valid even when eddies are generated.

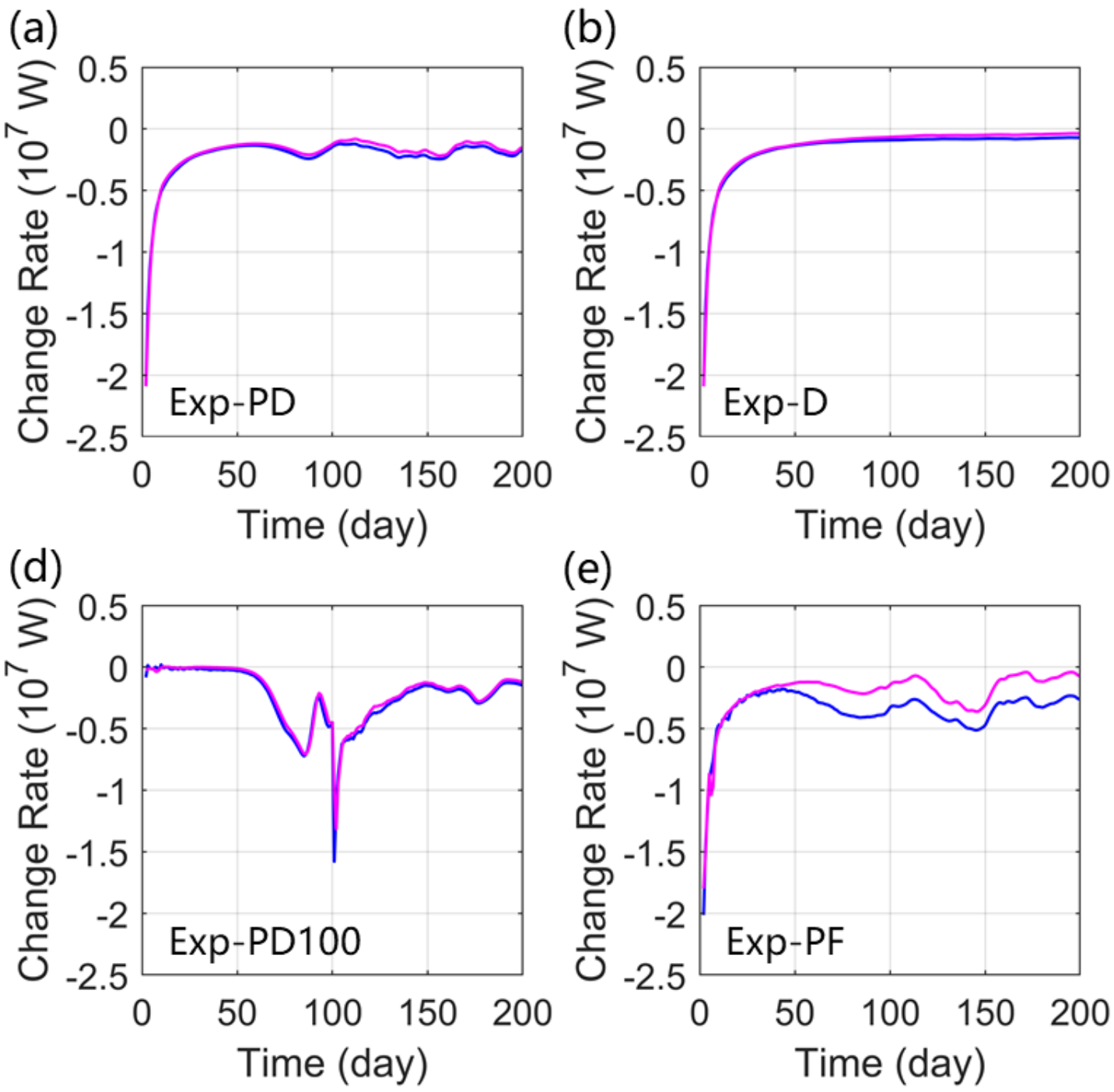

**Figure 12. The change rate of APE (magenta curve) and the vertical buoyancy flux $g(\rho - \rho_r)w$ (blue curve) integrated over the model domain as functions of time from the base and sensitivity experiments.**

Although both the ice friction and eddies can produce negative buoyancy flux, their results are different. The eddy-induced buoyancy flux means energy transfer from APE to EKE through baroclinic instability, so there is no need for the loss in mechanical energy. The ice friction, however, does negative work on the ocean via modifying the density and velocity structures (the isopycnal slope is reduced when the buoyancy flux is negative). In Exp-D, as the APE is released by the ice-induced buoyancy flux and the KE decreases due to Ekman pumping (Fig. 8, black dashed lines), the mechanical energy gets lost (Fig. 11b). The relative importance of the Ekman pumping versus baroclinic instability for the release of APE is indicated

by the difference between Exp-P (no Ekman pumping) and Exp-PF (with Ekman pumping). The EKE is significantly less in the Exp-PF (Fig. 9d), both near the surface and at 100 m, because more APE is released by Ekman pumping rather than being converted to EKE through baroclinic instability.

For existing eddies, the friction between the ice and eddies is spatially variable and thus gives rise to strong Ekman pumping (see the case in Fig. 13). As the ice-induced Ekman pumping penetrates into deep water, it enhances the vertical velocity and produces stronger buoyancy flux (Figs. 14b and d) compared to that induced only by eddies (Figs. 14a and c). Although the buoyancy flux is positive somewhere (Figs. 14c and d), the integral result over the model domain is negative, resulting in the net loss of APE (Figs. 12c and d). Note that this is essentially the same mechanism as for the mean flow and thus the sign of buoyancy flux is independent of the sense of vorticity of the eddy (cyclonic or anticyclonic), although the magnitude would change if we considered the relative vorticity in the Ekman transport calculation.

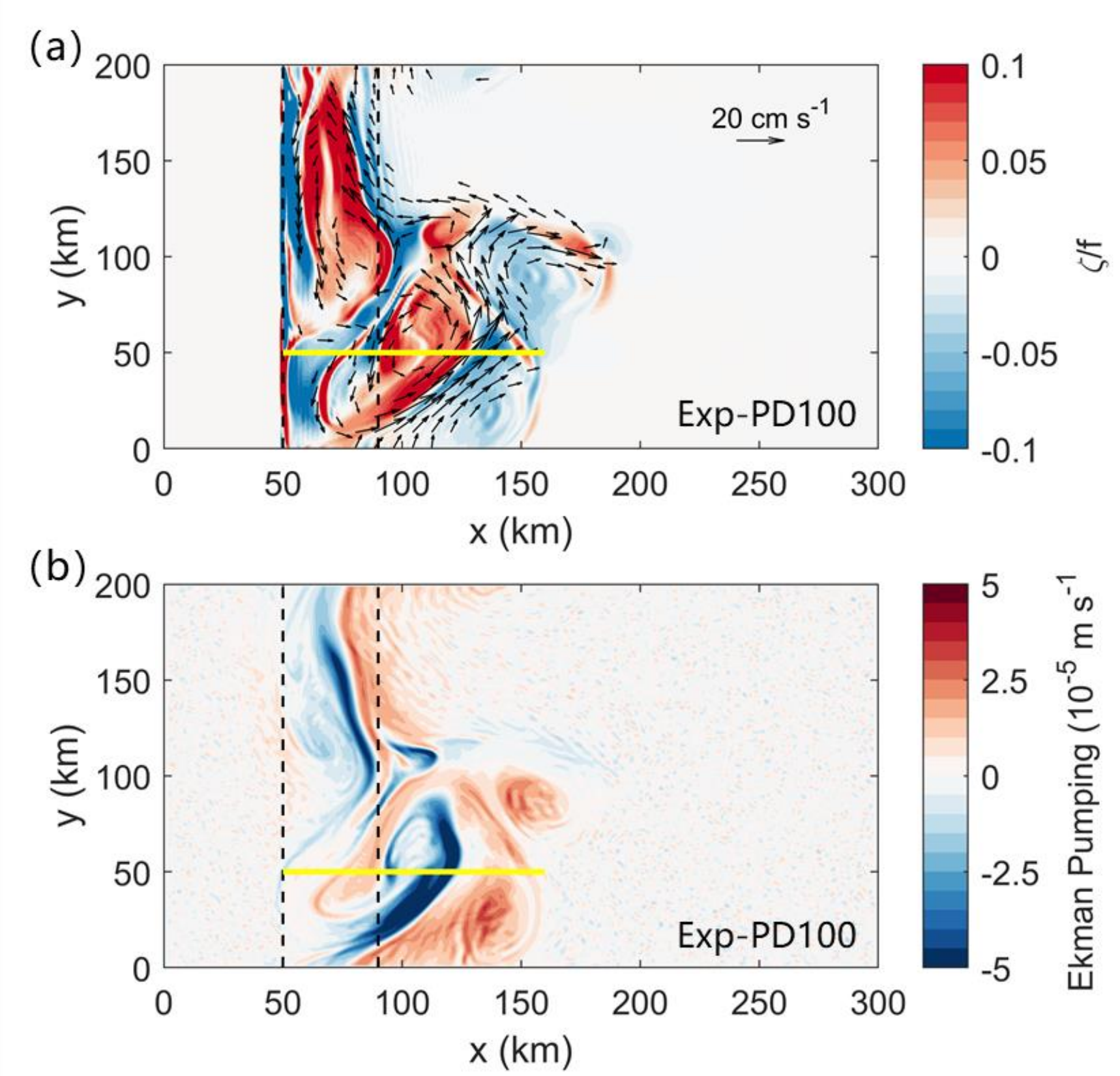

**Figure 13. Distributions of (a) the surface normalized relative vorticity $\zeta/f$ and velocity vectors faster than 5 cm s$^{-1}$ and (b) ice-induced Ekman pumping at *t* = 101 days from Exp-PD100. The yellow line marks a cross-channel section spanning from *x* = 50 to 110 km at *y* = 50 km.**

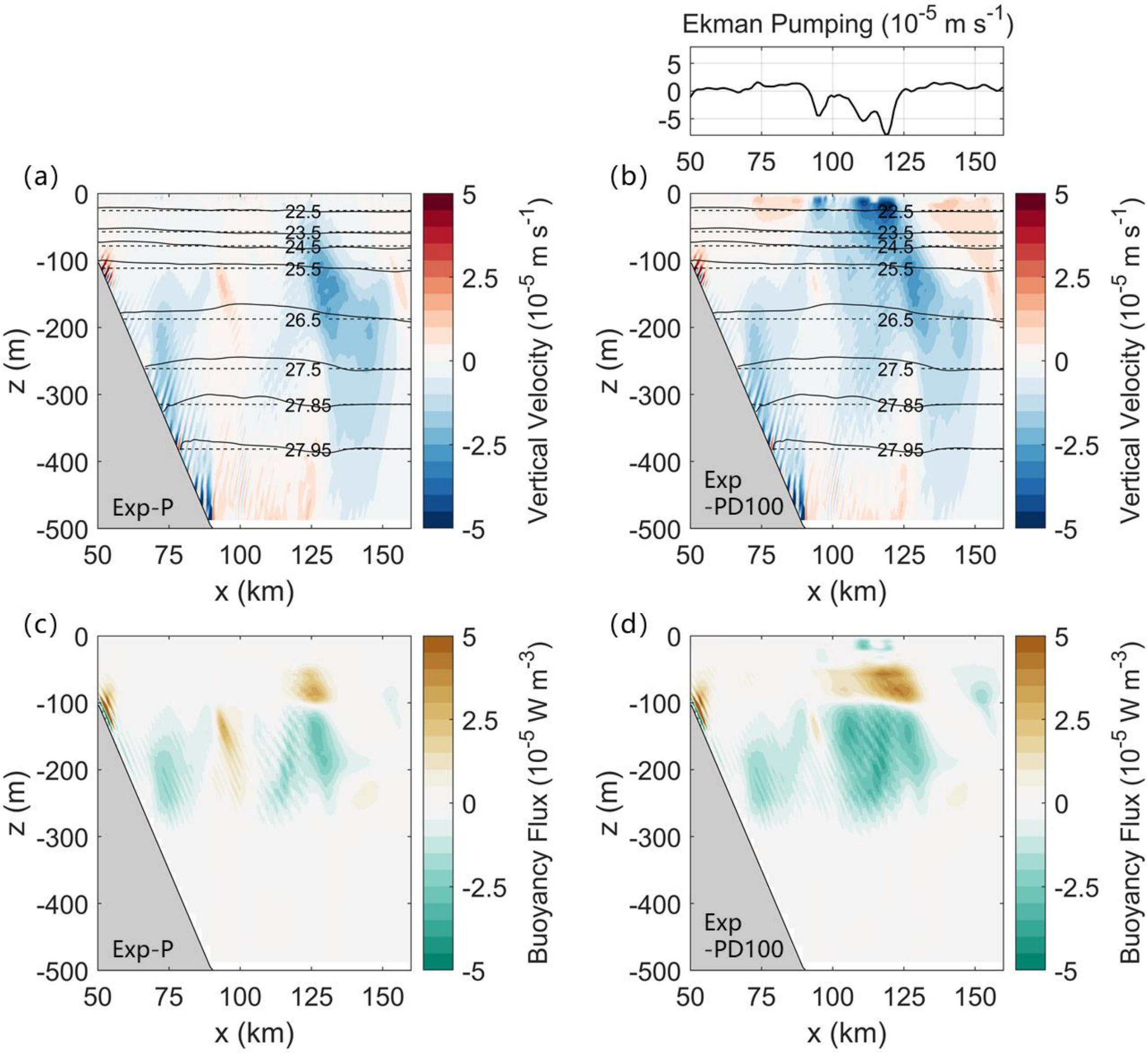

**Figure 14. Vertical sections of (a and b) the vertical velocity (color) overlain by isopycnals (black solid contours) and (c and d)**
335 **vertical buoyancy flux at $t$ = 101 days (see the location of the section in Fig. 13). The left column is from Exp-P and the right column is from Exp-PD100. The black dashed lines in (a and b) indicate the background mean isopycnals. Above panel (b) is the Ekman pumping at $t$ = 101 days from Exp-PD100.**

### 3.3 Scaling estimates for the change of APE

The results presented above indicate that both the ice-induced Ekman pumping and baroclinic instability have contributions to
340 the change of APE. Here we summarize some scales that may be useful to measure their relative importance.

The rate of conversion from mean APE to EKE (in units of W m$^{-2}$) through baroclinic instability can be calculated by (Gill et al., 1974; Smith, 2007)

$$R_{BC} = \frac{\rho_0 v_{max}^2}{2} \int_{-h}^{0} \frac{f^2}{N^2} \frac{d\theta}{dz} \left( \frac{|\hat{\psi}|}{|\hat{\psi}|_{max}} \right)^2 \frac{\mathbf{K}}{|\mathbf{K}|^2} \cdot \frac{d\mathbf{U}}{dz} dz \,, \tag{13}$$

where $v_{max}$ is the maximum eddy velocity, $\theta = \tan^{-1}(\hat{\psi}_i/\hat{\psi}_r)$ is the phase of $\hat{\psi}$ (note that the eigenvector can be written as $\hat{\psi} = |\hat{\psi}|e^{i\theta} = \hat{\psi}_r + i\hat{\psi}_i$), and $\mathbf{K}$ is the wavenumber vector. From Eq. (13), the baroclinic instability conversion rate may be scaled as $R_{BC} = \mathrm{O}(\rho_0 V_e^2 VIH/L^2)$, where $V_e$ is the order of eddy velocity, $V$ is the order of mean flow velocity, $I$ is the horizontal scale of the fastest-growing mode, $H = fL/N$ is the vertical scale of the mean stratification, and $L$ is the horizontal scale of the mean flow.

The ice-induced Ekman pumping helps release the APE at a rate of $g(\rho - \rho_r)w$, where $\rho - \rho_r = \mathrm{O}[\rho_0 fLV/(gH)]$ is derived from the thermal wind relation and $w = \mathrm{O}[C_{Di}V^2/(fL)]$ is of the same order as Ekman pumping (Leng et al., 2022). Integrating $g(\rho - \rho_r)w$ vertically gives the rate of change of APE (in units of W m$^{-2}$), $R_E = \mathrm{O}(\rho_0 C_{Di} V^3)$. The presence of $V^3$ in the scaling suggests that it is very sensitive to velocity and is likely to be more important around the boundaries of Arctic. With these scales, we obtain a ratio that measures the relative importance of baroclinic instability versus Ekman pumping in the release of APE

$$\epsilon = \frac{R_{BC}}{R_E} = \frac{IHV_e^2}{C_{Di}L^2V^2}. \tag{14}$$

If we take $I = L = 10$ km, $H = 100$ m, and $V_e = V = 0.1$ m s$^{-1}$, the ratio $\epsilon$ is of order 1, suggesting that both the Ekman pumping and baroclinic instability are important. For larger scale mean flows ($L > 10$ km), the Ekman pumping will dominate over baroclinic instability. In our base experiment (Exp-PD), eddies are not developed in the first 60 days and the release of APE is dominated by the Ekman pumping. Due to the growth of eddies and the weakening of mean flow, the effect of baroclinic instability on the change of APE becomes comparable to that of Ekman pumping.

## 4 Summary and discussion

We have explored the role of ice friction in the slope current energetics using an idealized primitive equation numerical model. Results from a set of experiments and scaling analysis show that the ice friction modifies the EKE in three distinct ways. First, ice friction acts to suppress the growth of surface intensified modes whose vertical scale is on the same order of magnitude as the thickness of the Ekman layer. Second, Ekman pumping releases APE of the slope current, which reduces the vertical shear and weakens subsequent baroclinic instability. Finally, Ekman pumping acting on mesoscale eddies formed by baroclinic instability spins them down through a similar release of APE. These latter two mechanisms are indirect in the sense that they take place outside the thin Ekman layer, where friction is weak. For typical parameters, the vertical scale of this damping effect is O(100 m). A year-long mooring observation on the Chukchi slope supports our numerical results and scaling analysis regarding the ice-induced damping of KE of the slope current and eddies (Fig. A1 in Appendix A). During warm months the

region is not fully ice-covered and the KE maximum is located at a depth shallower than –50 m. When the region is covered by packed ice, the KE above –150 m is significantly weakened and the KE maximum occurs around the –100 m depth.

The scaling analysis suggests that the ice-induced Ekman pumping will dominate the release of APE for large scale flows, but the baroclinic instability conversion is also important when the horizontal scale of the mean flow is on the order of the baroclinic deformation radius and the eddy velocity grows to be of the same order as the mean flow velocity. It is also worth noting that the ice-ocean drag can vary by over an order of magnitude and can be much larger under rough sea ice (e.g., Steiner et al., 1999). A roughness-dependent ice-ocean drag may be required for better estimating the ice-induced Ekman pumping and its contribution to the release of APE.

The presence of the continental slope is important in shaping the flow structure during the evolution driven by the ice friction. As revealed in Exp-PD, the ice-induced overturning reaches the bottom and sets up a counter current spanning from the shelfbreak to the open ocean, with the maximum velocity being trapped in the vicinity of the shelfbreak (Fig. 7c). This suggests that both the Chukchi Shelfbreak Jet and the Atlantic Water Boundary Current are likely to be affected by the ice friction. In the absence of a continental slope, the analytical and numerical models of Leng et al. (2022) captured a similar counter current, but it did not show a bottom-trapped feature. The bottom friction also affects the energetics of the mean flow and eddies. When we turn off the bottom drag, the deep current becomes stronger and more eddies are generated (not shown).

Over the past decades, there have been significant changes in the ice condition across the Chukchi Sea and surrounding areas, including the reduction of sea ice extent and the transition towards longer ice-free season (Frey et al., 2015). If these changes continue, more eddies would be generated and survive for longer periods to drive exchanges of water and energy between the CSC and BG (the ice friction becomes of less importance). Further work is required to explore the interaction between the CSC and BG with consideration of the changing ice condition.

**Appendix A**

Figure A1 shows the satellite-based sea ice concentration and mooring-observed KE on the Chukchi slope from October 2013 to September 2014. The ice concentration product is the Climate Data Record of Passive Microwave Sea Ice Concentration (version 3) from the National Snow and Ice Data Center (NSIDC), which has a temporal resolution of one day and spatial resolution of 25 km (Peng et al., 2013; Meier et al., 2017). The mooring data were provided by the Bureau of Ocean and Energy Management (BOEM). We calculate the daily mean KE (in units of $m^2\ s^{-2}$) using the velocity data at moorings CS4 and CS5 (vertical resolution of 5–10 m). These two moorings were located near the center of the slope current. Details of the data description and processing have been given by Li et al. (2019).

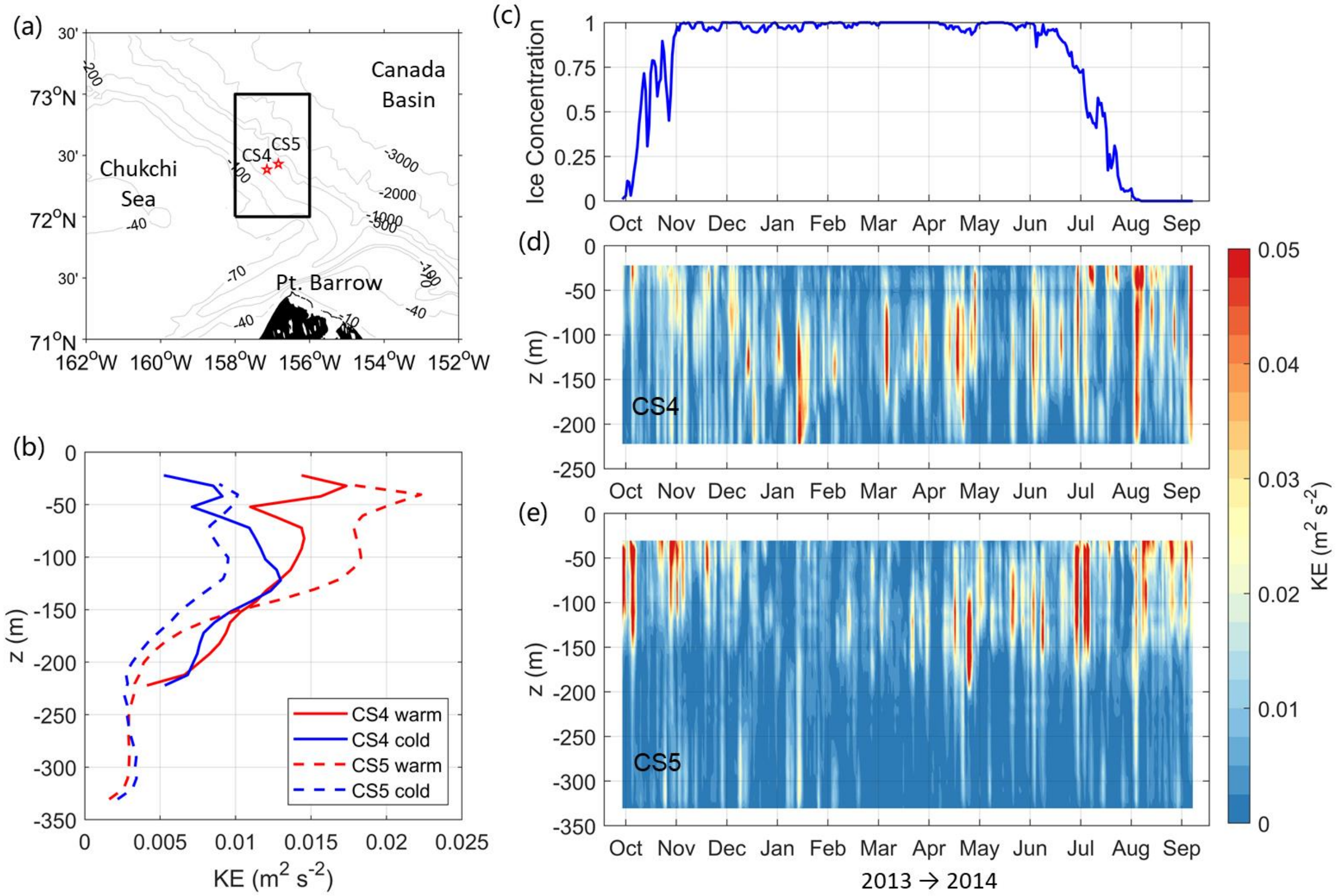

400 **Figure A1. (a) Locations of the BOEM Chukchi slope moorings CS4 and CS5. The black box outlines the region for calculating the ice concentration. (b) Vertical profiles of the KE averaged over warm (October and November 2013 and July to September 2014) and cold (December 2013 to June 2014) months at moorings CS4 and CS5. The warm (cold) month is defined as the month with the mean ice concentration less (larger) than 0.9. (c) Time series of the area-mean ice concentration from October 2013 to September 2014. (d) and (e) Time-depth plots of the KE at moorings CS4 and CS5.**

## Data Availability

The 2005-17 World Ocean Atlas climatology is available from the National Ocean Data Center: https://www.ncei.noaa.gov/access/world-ocean-atlas-2018/bin/woa18.pl. The NSIDC sea ice concentration data are available at https://doi.org/10.7265/N59P2ZTG. The Chukchi slope mooring data are available through the Bureau of Ocean and Energy Management (https://www.boem.gov). The numerical model configuration, parameters, and forcing fields are stored at https://zenodo.org/record/7317884#.

## Author contributions

HL designed the model experiments, generated figures, and drafted the manuscript. HH and MAS provided suggestions and contributed to manuscript revision.

## Competing interests

The contact author has declared that neither of the authors has any competing interests.

## Acknowledgments

This work was funded by the National Natural Science Foundation of China (Grant 42006037) and the Chinese Polar Environmental Comprehensive Investigation & Assessment Programs. MAS was supported by the National Science Foundation (Grants OCE-2122633 and OPP-2211691).

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
