# Peer review of "A numerical investigation on the energetics of a current along an icecovered continental slope"

_EGUsphere, 2022_

## Author Comment (AC1)

**Referee's comments are in black text, authors' responses are in blue.**

**Review of "A numerical investigation on the energetics of a current along an ice-covered continental slope" by Leng et al.**

In this manuscript the authors build on recent work exploring the impact of sea ice cover on ocean dynamics and baroclinic instability. The simulations are well chosen, the theoretical work is generally clear, and the results are compelling.

However, I think the manuscript would be easier to read if it were slightly restructured. I also have a few minor suggestions.

Edward Doddridge

Thank you very much for the comments and helpful suggestions. We have restructured the manuscript and addressed all comments. Below are our point-by-point responses. The line numbers mentioned below refer to the lines in the tracked-changes version of the manuscript.

**Comments**

**Structure and story**

The manuscript contains a lot of great science, however, it is not as easy to read as it could be. The names of the control and sensitivity experiments are all very similar, and the current structure requires readers to remember all of the different simulations and use that knowledge while reading all of the paper. The paper would be easier to digest if the sensitivity experiments were introduced in section 3.2 when they are discussed.

Thank you for the suggestion. We have moved the introduction of the sensitivity experiments to section 3.2. This does make the paper easier to read.

**Eddy spin down**

There is a wealth of previous work examining the impact of surface stress on mesoscale eddies outside of the sea ice zone. The manuscript would be strengthened by engaging with this literature, for example Munday et al. (2021) and Seo et al. (2019), and the references within. In particular, the discussion in lines 366-367 would benefit from this addition.

Thank you for sharing these papers. We have discussed the spin-down of eddies caused by relative wind stress and compared this with the damping effect of ice friction (Lines 473-477).

**Minor comments**

Line 85: The description of the initial velocity state would be clearer if equations 1a) and 1b) were swapped. As written, the x dependence of the initial velocity field is not immediately obvious – I spent longer than I care to admit looking for an x in the right hand side of 1a).

We have simplified equation 1 to clearly show that the initial velocity is x- and z-dependent (Line 88).

Lines 104-105: positive downward radiation would act to melt the ice, not maintain it.

Sorry for the misleading wording, we have rephrased the sentence and mentioned that those are representative winter-time downward radiations in the Arctic (Lines 112-115). The ice melting is very weak in our experiments, so the model domain is covered by packed ice throughout the simulation.

Line 136: This should be rho_0 to be consistent with the Boussinesq approximation used by MITgcm. E.g Nycander (2011).

We have replaced "rho" with "rho_0" in the calculation of KE (Line 150) and updated all figures which show the results of KE. The new calculated KE is almost the same as our old calculation.

Line 149: why is the power from the ice friction an estimate? These variables can be directly obtained from the model and power calculated exactly.

We have replaced "estimated" with "calculated" (Line 164) since we calculated the power from the model output variables exactly.

Line 161-162: A statement regarding the magnitude of the relative vorticity would help justify ignoring the relative vorticity of the mean flow.

We have provided a scaling analysis to show that the relative vorticity of the mean flow is small and can be neglected (Lines 175-177). Note also that we calculate the PV gradient at the center of the front, where the background relative vorticity is zero.

Lines 239-243: This paragraph is poorly phrased. The phrase 'steady state' is used to refer to the evolving state prior to the generation of eddies – this is not a steady state since the flow and density surfaces are evolving. Only in an actual steady state would the intersection of streamlines and density surfaces require a diapycnal transport.

We admit it is confusing to refer to the evolving state prior to the generation of eddies as the 'steady state'. We have rephrased the paragraph to illustrate that the ice-induced overturning does not drive significant diapycnal transport (Lines 272-277).

Line 281: "maintains"? Should probably be 'remains' or 'is'.

We have replaced "maintains" with "is" (Line 345).

Lines 288-289: Does interior friction refer to viscosity?

Yes, we have mentioned this in Line 350.

Figure 11e): It may be a plotting issue, but it looks as though the work done by the surface stress is larger than the reduction in mechanical energy at the start of this panel.

Thank you for pointing out this issue. We plotted the two curves (in each panel of Fig. 11) starting from day 0, which is OK for the change of mechanical energy since it is relative to the initial condition (the change is zero on day 0). However, the work done by the surface stress is calculated from daily output variables starting from day 1, so the red curve should start from day 1 rather than day 0. We have moved the red curve to the right by a one-day time step. The updated Fig. 11 shows that the work done by the surface stress is smaller than the reduction in mechanical energy.

From day 100 onwards, it looks as though the ice-ocean stress is putting a small amount of energy back into the ocean. What is going on here? Has the mean current reversed?

Exp-PF is forced by the surface stress diagnosed from Exp-PD. Although this stress spins down the mean flow, it does not always damp surface eddies as in Exp-PD. When $\boldsymbol{\tau} \cdot \mathbf{u}_s > 0$ (see the case in the figure below), the surface stress can do positive work on surface eddies and put a small amount of energy back into the ocean as shown in Fig. 11e. We have mentioned this in Lines 356-358.

[Figure]

Exp-PF has produced counter currents similar to those in Exp-PD, which are set up by the Ekman pumping as described in Lines 263-267.

[Figure]

Lines 355-360: The figures for mechanical energy are very instructive. Can similar time series be constructed for the APE? This would explicitly show the changing importance of Ekman pumping and baroclinic instability.

Thank you for the suggestion. We have added the timeseries for the change of APE in Fig. 11. This does help illustrate the changing importance of Ekman pumping and baroclinic instability (see the discussion in Lines 440-444).

Lines 366-367: Discussion of previous work on eddy spin down would be appropriate here

Thank you for the suggestion. We think it's better to discuss the eddy spin down in a separate paragraph (Lines 472-477). This also leads to the following discussion about the effect of changing ice condition in the damping of eddies.

**References**

Munday, D. R., Zhai, X., Harle, J., Coward, A. C., & Nurser, A. J. G. (2021). Relative vs. Absolute wind stress in a circumpolar model of the Southern Ocean. Ocean Modelling, 168, 101891. https://doi.org/10.1016/j.ocemod.2021.101891

Nycander, J. (2011). Energy Conversion, Mixing Energy, and Neutral Surfaces with a Nonlinear Equation of State. Journal of Physical Oceanography, 41(1), 28–41. https://doi.org/10.1175/2010JPO4250.1

Seo, H., Subramanian, A. C., Song, H., & Chowdary, J. S. (2019). Coupled effects of ocean current on wind stress in the Bay of Bengal: Eddy energetics and upper ocean stratification. Deep Sea Research Part II: Topical Studies in Oceanography, 168, 104617. https://doi.org/10.1016/j.dsr2.2019.07.005

---

## Author Comment (AC2)

**Referee's comments are in black text, authors' responses are in blue.**

The manuscript by Leng, He and Spall looks into what releases Available Potential Energy (APE) in the Chukchi Slope Current. They build on an earlier paper by Leng, Spall and Bai (LSB) which pointed to how ice-ocean friction impacted the current. In this new manuscript the authors primarily compare the importance of mean friction-induced overturning to eddy overturning (by baroclinic instability) in reducing APE. They use idealized numerical model simulations to study the fully nonlinear adjustment in a set of spin-down experiments. They then conduct linear 1D quasi-geostrophic (QG) stability calculations to assess the baroclinic instability properties of the flow. The conclusions are that the large-scale frictionally-driven overturning is at least as large as the eddy-driven overturning in releasing APE.

I find that the study will make a useful contribution to our understanding of Arctic Ocean dynamics and, particularly, of how mesoscale eddies and sea ice impact the circulation. The study is for the most part well conducted and well written, so I will recommend that the paper is eventually published. There are nonetheless several issues that I would like the authors to address, both scientific and stylistic. I consider none of these to be crucial. But there are quite a few of them, and for this reason I will suggest that a 'major revision' is needed.

Thank you very much for the comprehensive reviews and helpful suggestions. We are very encouraged to see that our results are useful for understanding Arctic Ocean dynamics. We have revised the paper and addressed all scientific and stylistic issues. Below are our point-by-point responses. The line numbers mentioned below refer to the lines in the tracked-changes version of the manuscript.

In the following I will address the authors directly:

(l 67): It is claimed that the resolves mesoscale eddies. Here I expect you to define what you mean by 'mesoscale'. And if you mean that the first internal deformation radius is resolved well, then you'll need to report on how large this is, ideally both in the real Arctic and in your model.

In our model the mesoscale eddies are referred to as the eddies with a typical length scale of 10 km. This is also the scale of internal deformation radius in our model (10 km) and is in consistent with the first baroclinic deformation radius in the Arctic Ocean (from ~5 km in the Nansen Basin to ~15 km in the central Canada Basin). We have clarified this in Lines 70, 106-110.

2: Is these pure spin-down experiments? Please clarify.

The base experiment (Exp-PD) and three of the sensitivity experiments (Exp-D, Exp-P, and Exp-PD100) are pure spin-down experiments. The last sensitivity experiment (Exp-PF) is forced by the surface stress diagnosed from the daily output of Exp-PD. Although this stress also spins down the mean flow, it can do either positive (when $\boldsymbol{\tau} \cdot \mathbf{u}_s > 0$) or negative work (when $\boldsymbol{\tau} \cdot \mathbf{u}_s < 0$) on surface eddies, so Exp-PF is not a pure spin-down experiment. We have clarified this in Lines 134, 281-283.

(l 103): How is a downward radiative flux maintaining a sea ice cover? And does this buoyancy forcing imply that the model is in fact forced (so not pure spin-down experiments)? I also note that the surface mixed layer depth is kept to a minimum. How does this then contribute to the overall forcing of the model?

Sorry for the misleading wording. A downward radiation actually acts to melt sea ice, but the ice melting is weak for the given downward radiations as they are representative winter-time values in the Arctic. We have rephrased the sentences (Lines 112-115). Note that most of the radiations are reflected back to the air by sea ice and the buoyancy flux (related to ice melting) is very small and should not affect the forcing of the model. We prescribed a minimum surface mixed layer depth of 10 m just because it is close to the observed winter mixed layer depth in the Arctic. We have restarted Exp-PD without prescribing a mixed layer depth and found that the output mixed layer depth is only about 3 m. The KE and APE from the restart calculation are nearly identical to the previous results, so we don't think the forcing and energetics of the model are very sensitive to the prescribed minimum surface mixed layer depth.

(l 143 + eqn. 6): Here (in the definitions of KE) only v (north-south) is used. Is this because the expressions pertain to the mean flow? The north-south-averaged u should be small but not necessarily zero at every instance in time. Please clarify/discuss.

The mean flow here is referred to as the flow along the slope (north-south), so the cross-channel velocity u can be classified into perturbation velocity and should not be included in the calculation of mean flow KE. We have clarified this in Lines 160-161.

(l162): Background relative vorticity is ignored in the 1D QG calculations, as it needs to be. But I would like to see some rough scaling showing that this is a safe assumption. The reader might wonder since the relative vorticity of the mean flow is central to the other aspect of the dynamics studied here, namely the uneven surface Ekman pumping.

We have provided a scaling analysis to show that the relative vorticity of the mean flow is small and can be neglected (Lines 175-177). Note also that we calculate the PV gradient at the center of the front, where the background relative vorticity is zero.

6: In the stability calculation, only north-south wave propagation is accounted for (k=0). Admittedly, these are likely the fastest-growing waves. But please discuss this briefly.

Yes, the waves traveling in the north-south direction can most easily draw energy from the background mean flow since the background PV gradient is in the west-east direction. We have discussed this in Lines 187-188.

7: The instability machinery assumes QG. It then makes no sense to study wavelengths down to 1 meters, as done here. A point should be made of this, i.e. that one cannot assume QG to be valid down to this range.

We have mentioned that the results for length scales down to O(1 m) are not valid (Lines 225-226).

8: You might try to study growth in log scales in Fig. 5 to investigate whether you observe an exponential growth stage at any time. This might help a bit with the discussion on the bottom of pg. 5 and top of pg. 6. Just a suggestion.

Thank you for the suggestion. We have added the plot of the natural log of EKE to Fig. 5. The curve shows that the perturbations have started to grow around day 30, but the EKE is not large enough to be seen until day 60. The slope of that curve represents the exponential growth rate of EKE, which is ~0.1 per day for the growth period from day 30 to 90. We have discussed this in Lines 205-207, 230-233.

9: In Fig. 6b, would the fit between EKE and the eigenvector improve at all if you plotted the square of the eigenvector (as energy is a squared quantity)? In other words, to get at EKE profile from the linear modes, you would have to square.

We have updated Fig. 6b to show the square of the eigenvector. The normalized EKE fits well with the square of the eigenvector in the upper 100 m, so the ice-induced Ekman pumping is not important here (the Ekman pumping diminishes quickly along with the decay of the surface flow and has been very weak after day 60). The EKE below 100 m decays more slowly towards the bottom than the square of the eigenvector. This is likely related to the nonlinear evolution of eddies, which is not represented by the linear instability mode. We have rephrased the comparison between EKE and the square of the eigenvector (Lines 227-229).

(l 229-32): The explanation for the emergence of the subsurface depth could be clarified. How would this overturning remove the surface maximum?

We have explained the formation of the subsurface velocity core in detail in Lines 257-263.

(l 240-42): There is no steady-state ever in these simulations, right? But really my problem is with the sentence "...the surface velocity and overturning are very weak and there are no vertical...". Please check and clarify.

We admit it is confusing to refer to the evolving state prior to the generation of eddies as the 'steady state'. We have rephrased the paragraph to illustrate that the ice-induced overturning does not drive significant diapycnal transport (Lines 272-277).

(l 253-54): Here it is argued that the halocline mode extends to the surface. But earlier you claim that this mode is sheltered from the ice friction. How do these statements relate?

The vorticity field at the surface is dominated by small-scale perturbations but it also presents mesoscale features similar to those in the mid-depth (compare Figs. 10c and d, from Exp-P). This is due to the fact that the halocline mode extends to the surface, although its amplitude at the surface is very small (Fig. 6b). We have mentioned this in Lines 304-307.

In the earlier section of instability analysis, we mentioned that varying the ice friction can cause a change in the vertical structure of the halocline mode. However, this effect is too weak that it cannot prevent the growth of halocline mode. Namely, the development of halocline eddies is "sheltered" from the ice friction.

(l 256-57): In Exp-PD100 the subsurface KE maximum at ~100m depth is also clearly reduced immediately after introducing friction at day 100 - even if less so than higher up. So you may want to modify/qualify your sentence discussing this experiment here.

We have rephrased the sentence to say that the reduction of KE in Exp-PD100 occurs at all depths and is more significant in the upper 60 m (Lines 315-316).

(l260-61): How is the large-scale Ekman pumping (releasing APE) a source of EKE? Clarify.

We mean that the large-scale Ekman pumping releases APE and thus reduces the source of EKE. Much of APE is released by the Ekman pumping instead of being converted to EKE through baroclinic instability, so Exp-PF produces lower EKE than Exp-P. We have clarified this in Lines 322-324.

15: In your discussion of Fig. 8 you focus first on KE (top panel) and then on APE (lower panel). The time-evolution of both quantities are of course intimately related. I, as a reader, would prefer that you discuss the time evolution of KE and APE simultaneously - as you go through experiment by experiment. An extra note: for the APE discussion there are at least a couple of experiments (PD & PD100) that you don't discuss. Make sure you mention at least something about all lines in these plots.

Thank you for the suggestion. We have rephrased the section and discussed the results of KE and APE simultaneously. All experiments have been mentioned (section 3.2.1).

(l265-67): The APE in Exp-P continues to decrease also after day 100, i.e. after the KE field has plateaued. Discuss.

Note that the KE is also being dissipated by the interior viscosity and bottom friction. Although the APE continues to decrease to the end of Exp-P, the loss of KE due to dissipation is of the same magnitude as the energy conversion from APE to KE after day 100, so the KE field plateaus. We have discussed this in Lines 310-313.

(l 269): How about defining total mechanical energy (you use it later as well) as "the sum of APE and KE"?

We have defined total mechanical energy as the sum of APE and KE (Line 350).

(l 270): What is 2-dimensional about the flow without eddies?

The initial perturbation is removed, so the flow is 2-dimensional (cross-channel velocity is zero) and eddies cannot be generated. We have clarified this in Lines 298-300.

(l 271-72): The APE of Exp-PF is *slightly lower* than in Exp-PD after eddies form. This is likely for the same reason as for why KE in this experiment is higher, namely that eddies are not damped. Eddies also contribute to APE release and are, presumably, more efficient at this in Exp-PF.

Yes, we have mentioned that the eddies also contribute to APE release, so the APE in Exp-PF is slightly lower than in Exp-PD after eddies form (Lines 326-327).

20: Is Fig. 10 mentioned in the text at all? This is a big figure and deserves a few lines of mention.

We have mentioned Fig. 10 along with the discussion of KE and APE (Lines 306 and 326).

(Eq. 11): I would rather say "...which assumes that...". But, more importantly, you will need to convince me that Eq. 11 is a QG form of the density equation. The QG formulation normally includes horizontal advection by the geostrophic flow. As I understand things, it's only when the lateral flow is ageostrophic (e.g. after integrating zonally around a periodic channel) that it may scale to be small compared to vertical advection of the background vertical density gradient (as e.g. done in Vallis's book on pg. 387).

We admit that ignoring horizontal advection terms in the density equation is a more restrictive assumption than the QG approximation. We had thought that Eq. 11 is only valid for Exp-D, where the geostrophic flow is nearly two-dimensional ($u_g \approx 0$) and density has little change in the along-channel direction ($\partial \rho / \partial y \approx 0$), i.e., the horizontal advection by the geostrophic flow is small. However, further calculations show that the change of APE agrees well with the vertical density flux in other experiments (except for Exp-PF). This suggests that Eq. 11 is a good approximation even when eddies are generated. Therefore, it is appropriate to use the derived Eq. 12 to describe the effects of Ekman pumping and baroclinic instability on the release of APE.

We have revised the statement for Eq. 11 and mentioned that this is more restrictive than the QG assumption (Line 368).

(l 303...but also other places): You write that g(rho-rho_r)*w is the vertical buoyancy flux. But there is at least a sign inconsistency here and even (to be pedantic) a unit problem. Buoyancy is defined as b=-g(rho/rho_0). So I suggest you define a proper buoyancy flux or you remove the 'g' and call it a density flux (also later in the text).

Thank you for the suggestion. We decide to call (rho-rho_r)*w the vertical density flux.

(l 302-03): There is awkward wording here: upwelling...upward and downwelling...downward. Please rewrite.

We have rewritten the sentence as "… the lighter water (relative to the area-mean density) is advected upward or the denser water moves downward" (Lines 374-376).

(l 305, 312 and Fig. 12 caption): Related to my comment 22 above. A buoyancy flux which will release APE is positive, not negative. The density flux is downward.

We decide to call (rho-rho_r)*w the vertical density flux and have revised the statements about this throughout the paper.

(l 313-14): "...so there is no need for the loss in mechanical energy." I would disagree with such an interpretation. It's correct that b.c. instability produces EKE from APE (actually, in the Lorentz energy cycle the transfer goes via EAPE). But of course total mechanical energy is lost eventually there too, via friction (ice, bottom, internal). And, actually, even your large-scale flow probably needs to go through such a route. How can the APE which is released by the large-scale Ekman pumping eventually be dissipated? The only route to dissipation goes through KE. So I urge you to think through whether you need to rephrase some of these statements.

We agree that the increased KE through baroclinic instability is eventually dissipated via friction and the total mechanical energy is lost as well. We have rephrased the sentences (Lines 386-390).

(l 321-27): This discussion of frictionally-induced Ekman pumping tied to existing eddies is not very convincing. To say something convincing here, especially about the net impact on vertical buoyancy flux, I'd say you'll need to average over lots of eddy motion. You've already defined eddies in a Reynolds sense as the deviation from the along-channel (north-south) mean. Could you do a Reynolds flux calculation, with and without friction applied to the eddy field?

Thank you for the suggestion. We have defined the perturbation density $\rho'$ and vertical velocity $w'$ as the deviations from the along-channel mean and use $(\rho' - \rho_r)w'$ to represent the perturbation vertical density flux. The averaged $g(\rho' - \rho_r)w'$ over the model domain is $-9.4$ Wm$^{-2}$ in Exp-PD100, more than twice the value in Exp-P ($-4.1$Wm$^{-2}$). This demonstrates that the ice-induced Ekman pumping over eddies can cause a net loss of APE (Lines 397-402).

(Fig. 14a): Are the background isopycnals (dashed lines) actually as flat as they appear here? I don't see anything which would suggest a thermal wind shear here.

Sorry for the misleading wording. The dashed lines here indicate the area-mean density profile so they must be flat. We have mentioned this in Line 414.

(l 346): Should the expression for R_BC have a H^2/L^2 in it (instead of H/L^2)?

We have checked the scaling. It is clear that here should be H/L^2. See below for the details.

[Figure]

(l 357-59): Note that here you mention a scale transition of about 10 km. But in both the abstract and the Summary section you talk of the (internal) deformation radius. If you want to mention the deformation radius anywhere in the text, you'll need to build up the story around it somewhat. Is 10 km approx. equal to the def. radius in your simulations. And why would this scale be the important scale? Note, by the way, that halocline eddies need not take the scale of the def. radius since they do not extend throughout the entire water column. The instability producing halocline eddies is not pure Eady instability involving top and bottom edge waves. So it's not obvious that the classical def. radius is a relevant scaling parameter for the problem.

In our model the deformation radius is approximately 10 km. This is similar to the length scale of the fast-growing halocline mode from the instability analysis, and also the halocline eddies in our simulations. We have discussed this in the model description section (Lines 106-110).

We agree that these eddies do not derive from the entire water column, but our estimate for the deformation radius is also not based on the entire water column. We also do not think it is coincidence that they have a scale similar to that deformation radius. Other forms of baroclinic instability, like the Phillips model, also scale with the deformation radius, not just the Eady model, but we don't want to get into all of that in this paper.